**EMBO** *reports*

# Caspase cleavage of influenza A virus M2 disrupts M2-LC3 interaction and regulates virion production

Carmen Figueras-Novoa[1,2], Masato Akutsu [3,4,13], Daichi Murata[3,5,6,13], Anne Weston[7], Ming Jiang [8], Beatriz Montaner [1], Christelle Dubois[9], Avinash Shenoy [10,11✉] & Rupert Beale [1,12✉]

## Abstract

**Influenza A virus (IAV) Matrix 2 protein (M2) is an ion channel, required for efficient viral entry and egress. M2 interacts with the small ubiquitin-like LC3 protein through a cytoplasmic C-terminal LC3-interacting region (LIR). Here, we report that M2 is cleaved by caspases, abolishing the M2–LC3 interaction. A crystal structure of the M2 LIR in complex with LC3 indicates the caspase cleavage tetrapeptide motif ($_{82}$SAVD$_{85}$) is an unstructured linear motif that does not overlap with the LIR. IAV mutant expressing a permanently truncated M2, mimicking caspase cleavage, exhibit defects in M2 plasma membrane transport, viral filament formation, and virion production. Our results reveal a dynamic regulation of the M2–LC3 interaction by caspases. This highlights the role of host proteases in regulating IAV exit, relating virion production with host cell state.**

**Keywords** Influenza; M2; Caspase; Autophagy; LC3
**Subject Categories** Microbiology, Virology & Host Pathogen Interaction; Post-translational Modifications & Proteolysis; Structural Biology

## Introduction

Influenza A virus (IAV) is an important human and animal pathogen. As well as being a major pandemic threat, seasonal IAV results in an estimated 290,000–650,000 respiratory deaths per year (WHO, 2023). IAV is an eight-segment genome negative-strand RNA virus that belongs to the *Orthomyxoviridae* family (Taubenberger and Kash, 2010). IAV is pleiomorphic, capable of producing both filamentous and non-filamentous virions (Chu et al, 1949). Whereas non-filamentous virions are ~120 nm diameter spheroidal structures, filaments can be several microns in length. While pathogenic isolates of influenza produce a mixture of filamentous and spherical particles, laboratory-adapted strains are generally spherical. Virion shape is largely controlled by genomic segment 7, which encodes both the main structural determinant Matrix 1 (M1) and Matrix 2 (M2) proteins (Elleman and Barclay, 2004; Rossman et al, 2010). The IAV envelope is derived from the host cell plasma membrane, containing viral hemagglutinin (HA) and neuraminidase (NA) glycoproteins (Lamb, 2008). A small amount of M2 is also incorporated into virions. M2 is a single-pass membrane protein, comprising an external N-terminal domain, transmembrane domain, and a cytosolic C-terminal domain. M2 forms a tetrameric pH-regulated, proton-selective ion channel, activated by low exterior pH, and is a target for antiviral drugs (Holsinger et al, 1994; Pinto et al, 1992; Shimbo et al, 1996; Scott et al, 2020). The M2 protonophore activity is required for both viral entry and egress, where premature activation of HA would otherwise occur during trafficking through the secretory pathway (Takeuchi and Lamb, 1994; Alvarado-Facundo et al, 2015; Ciampor et al, 1992).

De-acidification of the secretory pathway results in erroneously neutral compartments. Erroneously neutral pH induces the recruitment of ATG16L1, a component of the core autophagy machinery. The E3 ubiquitin ligase-like ATG16L1 complex covalently lipidates ubiquitin-like ATG8 molecules, comprising the LC3 and GABARAP proteins. This conjugation of ATG8s to single membranes (CASM) is distinct from canonical macroautophagy (Beale et al, 2014; Ulferts et al, 2021). CASM depends on an interaction between the WD40 domain of ATG16L1 and the $V_1$H subunit of the V-ATPase, which is strongly inducible by M2 proton channel activity (Ulferts et al, 2021; Fletcher et al, 2018; Xu et al, 2019; Timimi et al, 2024).

Although CASM activation during IAV infection depends on M2 proton channel activity, M2 further modulates CASM by the presence of an LC3-interacting region (LIR) motif in its C-terminal domain (Beale et al, 2014). The M2 LIR motif directly binds to ATG8s and promotes their relocalisation to the plasma membrane. When point mutations are introduced in the M2 LIR, mutant viruses exhibit attenuated stability and reduced filamentous budding (Beale et al, 2014). The presence of a highly conserved LIR motif in M2 suggests that IAV not only triggers CASM as a cellular response, but also utilizes it during egress.

[1]Cell Biology of Infection Laboratory, The Francis Crick Institute, London, UK. [2]Faculty of Life Sciences, University College London, London, UK. [3]Buchmann Institute for Molecular Life Sciences, Institute of Biochemistry II, Goethe University, Max-von Laue-Str. 15, Frankfurt 60438, Germany. [4]Research Center for Advanced Analysis, National Agriculture and Food Research Organization (NARO), Tsukuba, Ibaraki, Japan. [5]Department of Biomolecular Chemistry, Kyoto Prefectural University, Hangi-cho, Shimogamo, Sakyo-ku, Kyoto 606-8522, Japan. [6]China Innovation Center, Shiseido China Co., Ltd., Shanghai, China. [7]Electron Microscopy STP, The Francis Crick Institute, London, UK. [8]High Throughput Screening STP, The Francis Crick Institute, London, UK. [9]Proteomics STP, The Francis Crick Institute, London, UK. [10]Department of Infectious Disease, Imperial College, London, UK. [11]Satellite Group Leader, The Francis Crick Institute, London, UK. [12]Division of Medicine, University College London, London, UK. [13]These authors contributed equally: Masato Akutsu, Daichi Murata. ✉E-mail: a.shenoy@imperial.ac.uk; rupert.beale@crick.ac.uk

Caspases, a family of cysteine proteases, are responsible for the execution of cell death pathways such as apoptosis and pyroptosis (Thornberry and Lazebnik, 1998). Caspases recognize four amino acid motifs that end in an aspartic acid, which is essential for substrate cleavage. Many caspase substrates have been identified, including viral and autophagy-related proteins (Puccini and Kumar, 2016; Richard and Tulasne, 2012). M2 has been proposed to be a target of caspase cleavage (Zhirnov and Klenk, 2009). We speculated that the role of caspase cleavage of M2 might be to remove the LIR motif in response to cellular caspase activation to modulate filamentous budding. Here we experimentally identify a caspase cleavage motif in the C-terminal domain of M2. Cleavage removes the LIR and disrupts the M2–LC3 interaction. Permanently truncated IAV M2 mutant virus was substantially attenuated and failed to form filaments, highlighting the importance of the M2–LC3 interaction and its modulation through cleavage.

## Results

### M2 is cleaved at $_{82}$SAVD$_{85}$ motif by caspases

To study M2 cleavage during infection, PMA-differentiated THP-1 macrophage-like cells were infected with IAV, and samples collected in 2-hour intervals between 8 and 16 hours (Fig. 1A). We observed the appearance of a faster migrating M2 which we hypothesized could correspond to an M2 cleavage product. We noted concomitant PARP-1 cleavage, indicating caspase activation and subsequent cleavage of substrates (Fig. 1A). To further analyze the role of caspases in M2 cleavage, THP-1 cells were infected with IAV and treated with Z-VAD-FMK, a pan-caspase inhibitor. We observed a lower molecular weight M2 band formed at late time points (Fig. 1A,B). However, the band corresponding to cleaved M2 was not present upon Z-VAD-FMK treatment, indicating that M2 is cleaved by caspases (Fig. 1B).

To identify the caspases responsible for M2 cleavage, we used HAP1 cells deficient for the three executioner caspases (caspase-3, −7, and −6) and initiator caspase, caspase-8 (McIlwain et al, 2013). We found that $\Delta CASP6$ cells had substantially reduced levels of cleaved M2 upon IAV infection (Figs. 1C and EV1A). To confirm the role of caspase-6 in M2 cleavage, $\Delta CASP6$ cells were reconstituted to stably express caspase-6 ($\Delta CASP6 +$ CASP6; Fig. EV1B). Expression of caspase-6 in $\Delta CASP6$ cells rescued M2 to cleavage to levels seen in the parental cell line (Fig. EV1B,C).

To further test the role of caspases in M2 cleavage, THP-1 cells were infected and subsequently treated with the caspase-6 inhibitor Z-VEID-FMK followed by immunoblots for M2 cleavage. Compared to vehicle-treated THP-1 cells infected with IAV, Z-VEID-FMK treatment significantly reduced the abundance of cleaved M2 (Figs. 1D and EV1D). This supported a role for caspase-6 in M2 cleavage. Caspase-3 (Z-DEVD-FMK) and caspase-8 (Z-IETD-FMK) inhibitors were also used to assess the roles of these caspases in M2 cleavage. Treatment with either inhibitor alone or in combination showed a partial, non-significant effect on M2 cleavage (Fig. EV1E,F). Taken together, these results point to a dominant role for caspase-6 in M2 cleavage during IAV infection and a partial role for caspase-3. We conclude M2 is cleaved predominantly by caspase-6 with some redundancy, and decided to further elucidate the caspase cleavage site and its consequences during IAV infection.

A caspase cleavage motif has been predicted in the C-terminal region of M2 (Zhirnov and Syrtzev, 2009). Because caspases have a strict preference for aspartic acid at the scission bond, we mutated three amino acids from aspartic acid to alanine (D85A, D87A, and D88A) to precisely identify the cleavage site (Martin, 2014). We found that substitution of aspartic acid 87 or 88 with alanine (M2$^{D87A}$ and M2$^{D88A}$ mutants) had no effect on the appearance of the cleaved M2 product on immunoblots (Fig. 1E,F). However, infection with IAV expressing M2$^{D85A}$ abolished the lower molecular weight M2 band, indicating inhibition of caspase cleavage (Figs. 1E,F and EV1G,H). To verify the tetrapeptide caspase-recognition motif is $_{82}$SAVD$_{85}$, we tested M2 cleavage in mutants including single amino acid changes upstream of D85 (M2$^{S82A}$ and M2$^{V84A}$). Both mutants were cleaved to a lesser extent than wild-type M2, indicating these amino acids promote substrate recognition and cleavage (Figs. 1E,F and EV1H). Together, these results show that during IAV infection, M2 is cleaved predominantly by caspase-6 and caspase-3, at the $_{82}$SAVD$\downarrow$A$_{86}$ motif.

We also wanted to confirm that caspase cleavage of M2 occurs in clinically important IAV strains. We therefore obtained MDCK-SIAT1 cells that had been infected by seasonal H3N2 IAV strains as part of the effort to match IAV vaccine antigenicity with circulating strains. Although as expected M2 expression levels differ between clinical isolates, M2 cleavage could be observed to varying degrees in every case (Fig. 1G). We noted variation in M2 amino acid sequence between strains, as seen in a multiple sequence alignment of the amino acids 77–97. Within this region, G89S was present in all strains (Fig. 1H). An S82N substitution was observed in 5 strains, which did not lead to a reduction in M2 cleavage. Notably, a V84A substitution (present in A/Uppsala/SE24-02917/2024) was associated with lower levels of cleaved M2 (Fig. 1G,H). This is consistent with the reduction in cleavage observed in the corresponding PR8 mutant, M2$^{V84A}$ (Fig. 1F). These results confirm that M2 cleavage is not limited to the laboratory-adapted PR8 strain, but also occurs in pathogenic IAV strains. We therefore decided to investigate the consequences of M2 cleavage, and lack thereof, during IAV infection.

### M2 cleavage disrupts M2–LC3 interaction

We asked whether cleavage of M2 at D85 had a functional consequence on the course of IAV infection. Downstream of the $_{82}$SAVD$\downarrow$A$_{86}$ motif, an LC3-interacting region (LIR, aa 91–94) has previously been described (Beale et al, 2014) (Fig. 1E). Therefore, we investigated whether M2 cleavage disrupted the M2–LC3 interaction.

Previous studies have reported that disruption of the M2 LIR motif reduces LC3B lipidation during infection (Beale et al, 2014). To analyze the role of M2 cleavage, we assessed LC3B lipidation induced by M2 whilst caspases were inhibited by Z-VAD-FMK. LC3B lipidation was significantly increased upon Z-VAD-FMK treatment, which correlated with the abrogation in the cleavage of M2 (Fig. 2A,B). These observations were consistent with caspase-mediated disruption of the M2–LC3 interaction.

To verify the results obtained with pharmacological inhibitors, we next used a genetic approach to generate IAV expressing truncated M2 (M2$^{\Delta86-97}$) (Figs. 1E and EV1G). We first assessed whether the M2$^{\Delta86-97}$ mutant, which lacks the LIR, was deficient in LC3B interaction through CoIP experiments (Fig. 2C). These

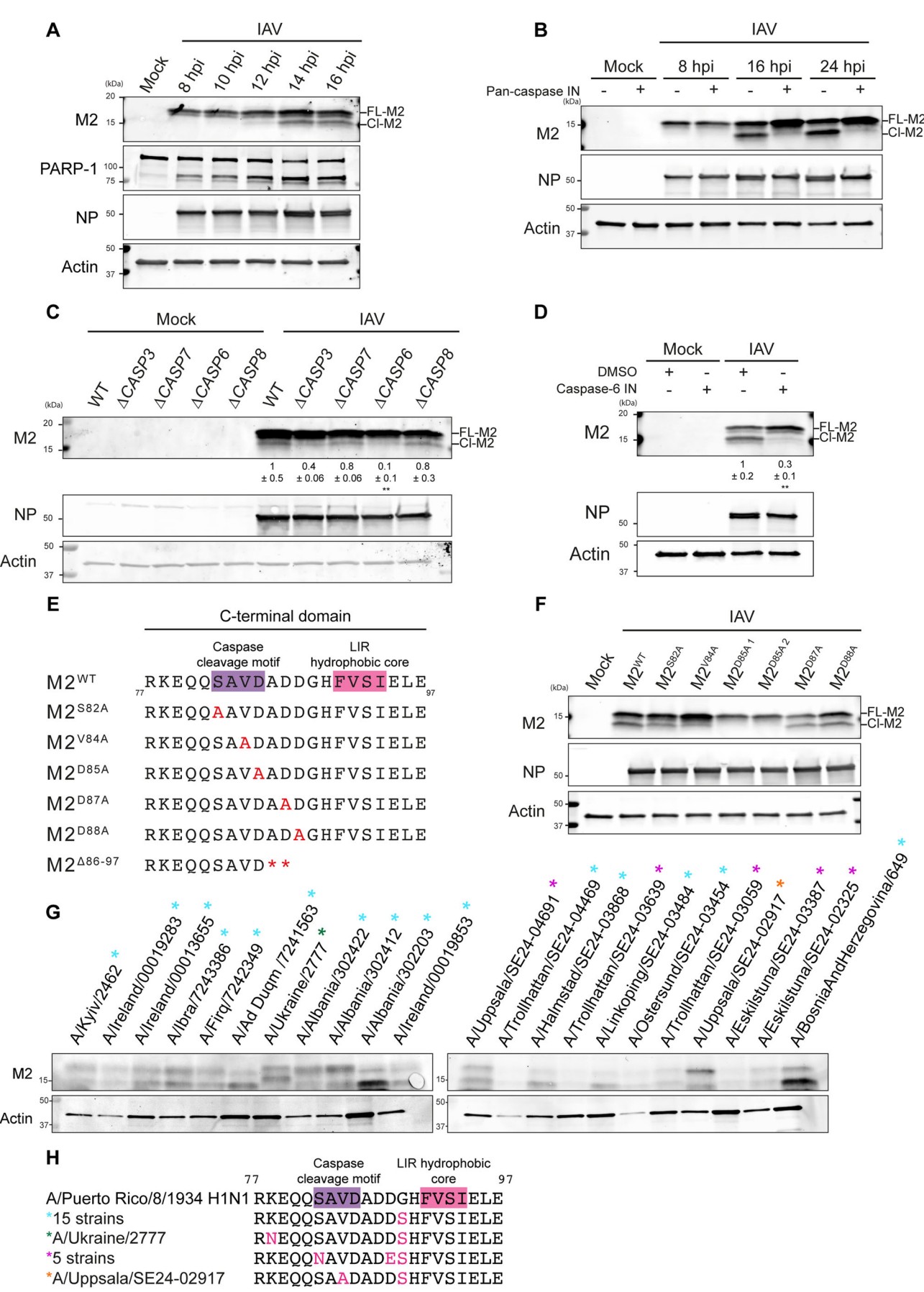

◄  **Figure 1.  M2 is cleaved at $_{82}$SAVD$_{85}$ motif by caspases.**

(A) Representative immunoblots of lysates of THP-1 cells infected with IAV PR8 for 8, 10, 12, 14, 16 h with an MOI of 10. (B) Representative immunoblots of lysates of THP-1 cells infected with IAV PR8 for the indicated duration with an MOI of 10. Indicated samples were treated with DMSO as a control, or 20 μM Z-VAD-FMK as indicated. (C) Representative immunoblots of lysates of wild-type (WT), ΔCASP3, ΔCASP7, ΔCASP6, and ΔCASP8 HAP1 cells. The indicated samples were from cells infected for 16 h with IAV PR8 at an MOI of 10. The ratio of cleaved M2 to full-length M2 was quantified. Mean ± SD of $n = 3$ biological replicates shown. **$P = 0.0065$. Ordinary one-way ANOVA with Bartlett's multiple comparisons. (D) Representative immunoblots of lysates of THP-1 cells infected with IAV PR8 for 24 h. Indicated samples were treated with DMSO as a control, or 20 μM caspase-6 inhibitor (Z-VEID-FMK) as indicated. The ratio of cleaved M2 to full-length M2 was quantified. Mean ± SD of $n = 3$ biological replicates shown. **$P = 0.0064$. Unpaired $t$ test. (E) Diagram depicting C-terminal region of IAV PR8 (strain A/Puerto Rico/8/1934). The caspase cleavage motif is highlighted in purple and the LIR hydrophobic core in pink. Amino acid changes for mutants mentioned are highlighted in red. (F) Representative immunoblots of lysates of THP-1 cells infected with IAV PR8 single amino acid mutants in and around the C-terminal caspase cleavage motif with an MOI of 10. (G) Immunoblots of lysates of MDCK-SIAT1 cells. Cells were infected with a range of seasonal IAV strains (H3N2) as indicated for 72 h. (H) Sequence alignment of amino acids 77–97 of IAV seasonal variants to the PR8 amino acid sequence, with caspase cleavage and LIR motif highlighted in purple and pink, respectively. A G89S was observed in 15 strains (A/Kyiv/2462/2024, A/Ireland/00019283/2024, A/Ireland/00013655/2024, A/Ibra/7243386/2024, A/Firq/7242349/2024, A/Ad Duqm/7241563/2024, A/Albania/302422/2024, A/Albania/302412/2024, A/Albania/302203/2024, A/Ireland/00019853/2024, A/Trollhattan/SE24-04469/2024, A/Halmstad/SE24-03868/2024, A/Linkoping/SE24-03484/2024, A/Ostersund/SE24-03454/2024, A/BosniaAndHerzegovina/649/2024), K78N and G89S substitutions were observed in the A/Ukraine/2777/2024 strain, S82N, D88E, and G89S substitutions were observed in five strains (A/Uppsala/SE24-04691/2024, A/Trollhattan/SE24-03639/2024, A/Trollhattan/SE24-03059/2024, A/Eskilstuna/SE24-03387/2024, A/Eskilstuna/SE24-02325/2024), and V84A and G89S substitutions were observed in the A/Uppsala/SE24-02917/2024 strain. Source data are available online for this figure.

experiments showed that LC3B pulled down M2 during IAV M2$^{WT}$ and M2$^{D85A}$ infection. However, this interaction was not present during M2$^{Δ86-97}$ infection (Fig. 2C). This finding established that the M2$^{Δ86-97}$ mutant is defective in LC3B interaction (Fig. 2C). Concordant results were obtained in CoIP experiments performed in THP-1 cells (Fig. EV2A). Taken together, these results indicate that the M2–LC3 interaction is mediated through the LIR, and it is specifically disrupted by caspase cleavage.

We then assessed LC3B lipidation levels in THP-1 cells infected for 24 h with IAV M2$^{WT}$, M2$^{D85A}$, and M2$^{Δ86-97}$ mutants (Fig. 2D,E). Consistent with M2 cleavage at D85, infection with the M2$^{D85A}$ mutant only presented full-length M2 on immunoblots, while only the shorter M2 fragment was present in M2$^{Δ86-97}$ infection (Fig. 2D). We observed M2$^{WT}$ and M2$^{D85A}$ viruses induced comparable amounts of LC3B lipidation, while there was a significant decrease in LC3B lipidation induced by the M2$^{Δ86-97}$ mutant (Fig. 2D,E). This points to a crucial role of caspase cleavage of M2 in modulating LC3B lipidation during infection. While LC3B lipidation during M2$^{Δ86-97}$ mutant infection is significantly decreased, residual LC3B lipidation is due to the proton channel activity of M2. This depends on the transmembrane region and remains intact in distal C-terminal domain mutants (Fig. 2D,E) (Nguyen et al, 2008; Schnell and Chou, 2008; Tobler et al, 1999; Ren et al, 2016).

To understand the interaction between M2, LC3B, and caspase cleavage at the molecular level, we obtained the co-crystal structure of the purified M2 C terminus (aa 70–97), containing the caspase cleavage site and LIR, and LC3B (aa 3–125). The structure was determined by molecular replacement using an LC3B structure (PDB: 2ZJD) as a search model (Fig. 2F). One complex pair of M2–LC3 is in an asymmetric unit where F91 and I94 at position 1 and 4 of M2 LIR fit into the typical hydrophobic pockets of LC3B (Fig. EV2B). The region of M2 upstream from the LIR (amino acids 70–85), which contains the caspase cleavage motif, exhibited no clear electron density in our crystal structure due to its flexibility. The electron density of the four amino acids preceding the M2 LIR hydrophobic core was clearly observed (Fig. 2G). We observed that the acidic side chain of D88 at −2 position of the M2 LIR forms two hydrogen bonds with the guanidinium group of R10 of LC3B, resulting in further stabilization of the M2–LC3 interaction (Fig. 2G). Consistent with the observed structural role of D87

and D88 for binding, we observed a decrease in LC3B lipidation when THP-1 cells were infected with M2$^{D87A}$ and M2$^{D88A}$ (Fig. EV2C). Collectively, these data clarify the role of acidic amino acids in the C-terminal region of M2. While D85 lies in a flexible region of M2 consistent with its role in caspase cleavage, D87 and D88 are important for stabilizing M2–LC3 binding.

These results show the distal C terminus of M2 possesses consecutive linear motifs, with the $_{82}$SAVD↓A$_{86}$ motif cleaving the hydrophobic core of the LIR motif (aa 91–94) and acidic amino acids upstream of it, which we have identified as relevant for the M2–LC3B interaction (Fig. 2H). Furthermore, analysis of 2189 unique M2 sequences in human IAV genomes showed high conservation of D85, consistent with this motif being important for viral fitness (Fig. 2I).

## Cleaved M2 is impaired in plasma membrane relocalisation

As M2 must travel to the plasma membrane for IAV budding, we assessed the impact of M2 cleavage on the plasma membrane localization of M2 (Wohlgemuth et al, 2018). To this end, we infected THP-1 cells with IAV M2$^{WT}$, M2$^{D85A}$, and M2$^{Δ86-97}$ mutants, and performed M2 immunofluorescence with and without permeabilization to assess internal and surface M2 abundance, respectively. The M2$^{WT}$ and M2$^{D85A}$ mutants showed similar internal localization, with a pool of M2 in the perinuclear region and substantial amounts at the plasma membrane (Fig. 3A,B). In M2$^{Δ86-97}$ mutant infection, M2 was still present in the perinuclear region, but localized less to the plasma membrane (Fig. 3A,B). We measured the number of M2-positive cells after surface staining, showing that an equivalent number of cells were infected by IAV M2$^{WT}$, M2$^{D85A}$, and M2$^{Δ86-97}$ (Fig. EV3A). To verify the reduction in M2 plasma membrane localization was due to the M2–LC3B interaction, an IAV PR8 mutant completely lacking the LIR motif (M2$^{FVSI\_AAAA}$) was generated. THP-1 cells were infected with M2$^{WT}$, M2$^{D85A}$, M2$^{Δ86-97}$, and M2$^{FVSI\_AAAA}$ for 8 h and assessed for M2 surface staining. While a comparable level of M2 intensity in the cell surface could be observed in M2$^{WT}$ and M2$^{D85A}$-infected cells, M2 plasma membrane intensity was significantly reduced in M2$^{Δ86-97}$ and M2$^{FVSI\_AAAA}$-infected cells (Fig. EV3B). These results

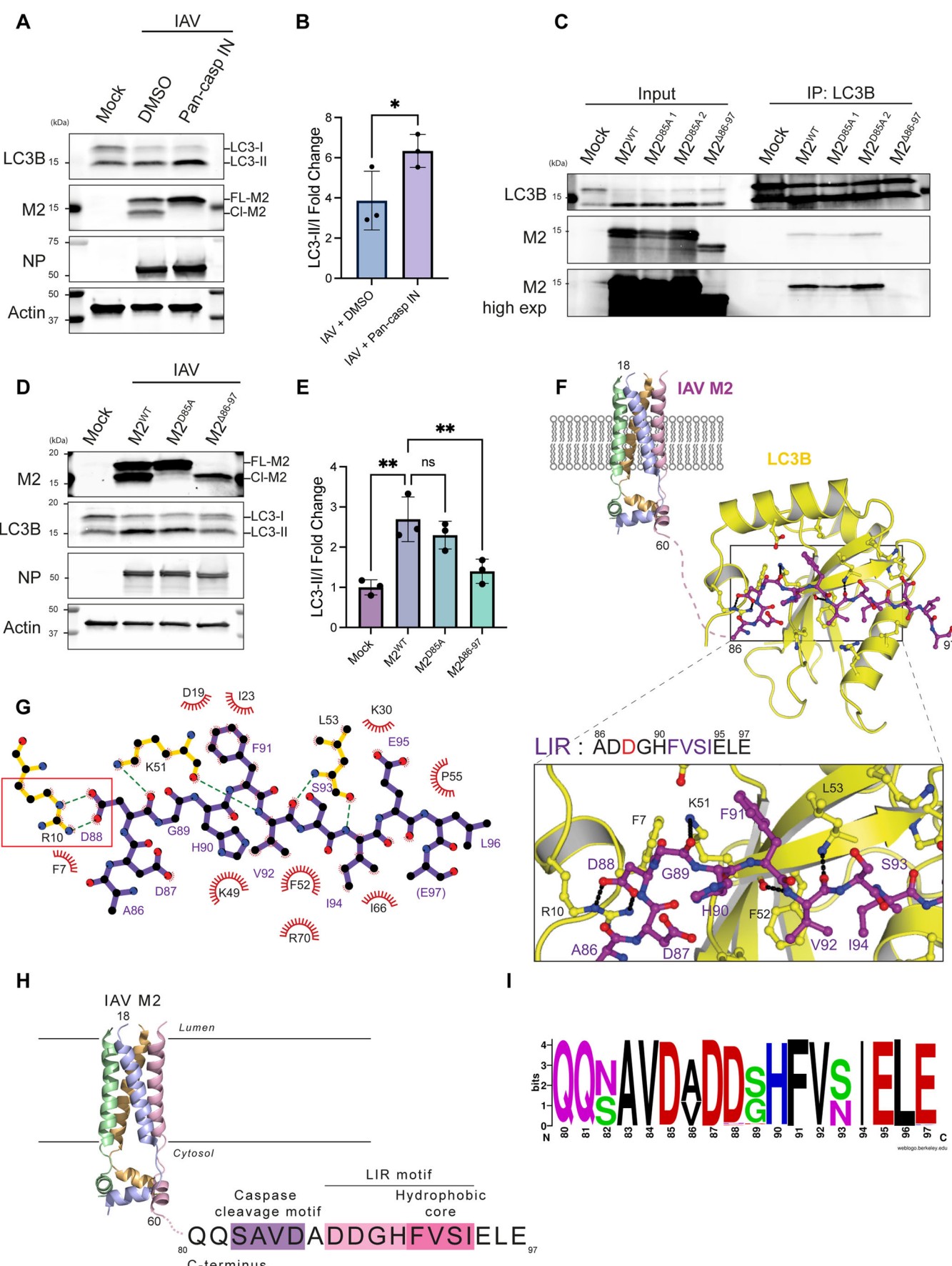

◀  **Figure 2.  M2 cleavage disrupts M2–LC3 interaction.**

(A) Representative immunoblots of lysates THP-1 cells infected with IAV PR8 for the indicated duration with an MOI of 10. After 8 h, indicated samples were treated with DMSO as a control, or 20 μM Z-VAD-FMK as indicated for a further 16 h. (B) Quantification of (A). Bars show mean ± SD of $n = 3$ biological replicates. *$P = 0.0357$. Ordinary one-way ANOVA with Dunnett's multiple comparisons. (C) Immunoprecipitation of endogenous LC3B from A549 cells analyzed by western blotting. Indicated samples were infected for 24 h with IAV PR8 M2$^{WT}$, or mutant strains, with an MOI of 10. (D) Representative immunoblots of lysates of THP-1 cells infected with IAV PR8 M2$^{WT}$, M2$^{D85A}$, and M2$^{Δ86-97}$ mutants for 24 h with an MOI of 10. (E) Quantification of (D). Bars show mean ± SD of $n = 3$ biological replicates. **M2$^{WT}$ vs. Mock $P = 0.0013$ and M2$^{WT}$ vs. M2$^{Δ86-97}$ $P = 0.0070$. Ordinary one-way ANOVA with Dunnett's multiple comparisons. (F) Model of IAV M2 proton channel (adapted from PDB 2RLF, aa 18–60) and LC3 in complex with M2 LIR. Crystal structure of LC3B in complex with M2 LIR: Ribbon and ball-and-stick representation model of the LC3B (Yellow) and M2 LIR (Purple, aa 86–97). (G) LIGPLOT representation of the interaction between LC3B and M2 LIR. M2 LIR is shown in purple. The hydrogen bonds are in green dashed lines (Wallace et al, 1995). (H) Diagram illustrating the IAV M2 proton channel (adapted from PDB 2RLF, aa 18–60). In the C-terminal region, the caspase cleavage motif (in purple) and preceded by the hydrophobic core (in pink) and acidic upstream amino acid (in light pink) of the LIR are highlighted. (I) Conservation analysis of amino acids 80–97 in C-terminal region of M2. From 54,974 human IAV M2 sequences, 2189 unique sequences were analyzed (Bao et al, 2008). Sequence logo was generated with WebLogo (Crooks et al, 2004). Source data are available online for this figure.

further suggest that the deficiency in M2 localization to the cell surface in truncation mutants is due to the loss of the LIR motif. Previously, an interaction between M2 and TRAPPC6A through L96 has been reported to mediate M2 transport to the plasma membrane (Zhu et al, 2017). To test the impact of this interaction on M2 transport on our system, a mutant deficient in this interaction (M2$^{EL–AA}$) was generated. While a decrease in M2 cell surface intensity could be observed with M2$^{Δ86-97}$ infection, no difference between M2$^{WT}$ and M2$^{EL–AA}$ could be observed (Fig. EV3C). Therefore, we conclude the last 11 amino acids of the M2 cytoplasmic tail regulate M2 plasma membrane transport.

It has been reported that the M2 LIR affects LC3B localization to the plasma membrane (Beale et al, 2014). To answer whether cleavage of the LIR motif affected M2 trafficking, we infected GFP-LC3B A549 cells with IAV M2$^{WT}$, M2$^{D85A}$, and M2$^{Δ86-97}$ mutants (Fig. 3C). In permeabilised cells, M2 was localized in the cytoplasm and around the plasma membrane region during infection (Fig. 3C). In mock-infected cells, a small number of GFP-LC3B spots were observed, reflecting basal levels of LC3B lipidation (Fig. EV3D). GFP-LC3B strongly colocalised with M2 during IAV M2$^{WT}$ and M2$^{D85A}$ infection, but not M2$^{Δ86-97}$ infection (Fig. 3C). To recapitulate our results from THP-1 cells, unpermeabilised GFP-LC3B A549 cells were stained with M2, and the intensity of M2 surface staining was quantified. We found M2$^{WT}$ and M2$^{D85A}$ mutants expressed similar amounts of M2 on the plasma membrane (Fig. 3D). However, the M2$^{Δ86-97}$ infection led to reduced amounts of M2 in the plasma membrane surface (Fig. 3D). We assessed GFP-LC3B spot number (Fig. 3E) and spot area (Fig. EV3E) in M2-positive cells, which confirmed that while M2$^{WT}$ and M2$^{D85A}$ infection were comparable, GFP-LC3B spot number and area were markedly reduced in M2$^{Δ86-97}$ infection (Figs. 3E and EV3E). To verify that the reduction in M2 plasma membrane localization and GFP-LC3B relocalisation is mediated through disruption of the M2–LC3B interaction, GFP-LC3B A549 cells were infected with M2$^{WT}$ and newly described LIR mutants M2$^{D87A}$ and M2$^{D88A}$, and stained for cell surface M2 expression. M2 surface staining and GFP-LC3B relocalisation was observed in M2$^{WT}$-infected cells (Fig. EV3F). However, infection with M2$^{D87A}$ and M2$^{D88A}$ resulted in decreased GFP-LC3B relocalisation, consistent with the importance of these residues for M2–LC3B binding (Fig. EV3F). In addition, reduced M2 surface staining was observed with the M2$^{D87A}$ and M2$^{D88A}$ relative to M2$^{WT}$ (Fig. EV3F), suggesting that the ability of M2 to interact with LC3B affects M2 transport to the plasma membrane.

## M2 cleavage reduces M2 incorporation into virions

To assess whether the decreased transport of M2 to the plasma membrane altered the composition and presence of different forms of M2 in virions, we purified IAV infectious particles from the supernatant of IAV M2$^{WT}$ infected MDCK cells at 48 hpi. Viral proteins could be identified at their expected molecular weights by Coomassie staining of purified infectious particles following SDS-PAGE (Fig. 4A). Cell lysate and infectious particles were run in parallel to assess ratios of full length to cleaved M2 present. While most M2 present in cell lysates at this time point was the cleaved form, there was a 1:1 mix of full length to truncated M2 in infectious particles (Fig. 4B,C). This suggests that full-length M2 may be preferentially incorporated into virions, which is consistent with its preferential plasma membrane localization.

We then proceeded to purify infectious particles from supernatant of M2$^{D85A}$ or M2$^{Δ86-97}$ infected MDCKs at 48 hpi (Fig. 4D). Coomassie staining of purified virions following SDS-PAGE showed the main structural IAV proteins were present at comparable levels in both mutants (Fig. 4D). We observed that the ratio of LC3II/LC3I in virions was reduced in M2$^{Δ86-97}$ viral particles when compared to M2$^{D85A}$, which suggests lipidated LC3B was preferentially incorporated into virions containing the M2 LIR motif (Figs. 4E,F and EV4A). We next performed proteomic analysis of purified M2$^{D85A}$ and M2$^{Δ86-97}$ virions, which indicated LC3B was more abundant in M2$^{D85A}$ virions (Fig. EV4B). The ratio of other viral proteins to M1 abundance was similar between the M2$^{D85A}$ and M2$^{Δ86-97}$ mutants. However, the overall abundance of viral proteins was increased in M2$^{D85A}$ purified infectious particles (Fig. EV4B,C). This is consistent with a deficiency in the budding of the M2$^{Δ86-97}$ mutant.

## IAV M2$^{Δ86-97}$ is attenuated

To assess whether the introduction of uncleavable or truncated M2 in IAV resulted in attenuated virus, we performed plaque assays to assess the titer (Plaque Forming Units (PFU)/mL) formed by each mutant. While M2$^{WT}$ and M2$^{D85A}$ had comparable titers, the M2$^{Δ86-97}$ mutant had a significantly decreased titer (Fig. 5A). This indicates that caspase cleavage of M2 produces an attenuated virus. Consistent with this, plaques produced by M2$^{Δ86-97}$ infection were smaller as compared to M2$^{WT}$ (Fig. 5B,C). The titer difference was recapitulated in THP-1 cells infected for 8 h with IAV M2$^{WT}$, M2$^{D85A}$, and M2$^{Δ86-97}$ mutants, which verified the reduction in M2$^{Δ86-97}$ titer (Fig. EV4D). To understand the stage of viral life cycle

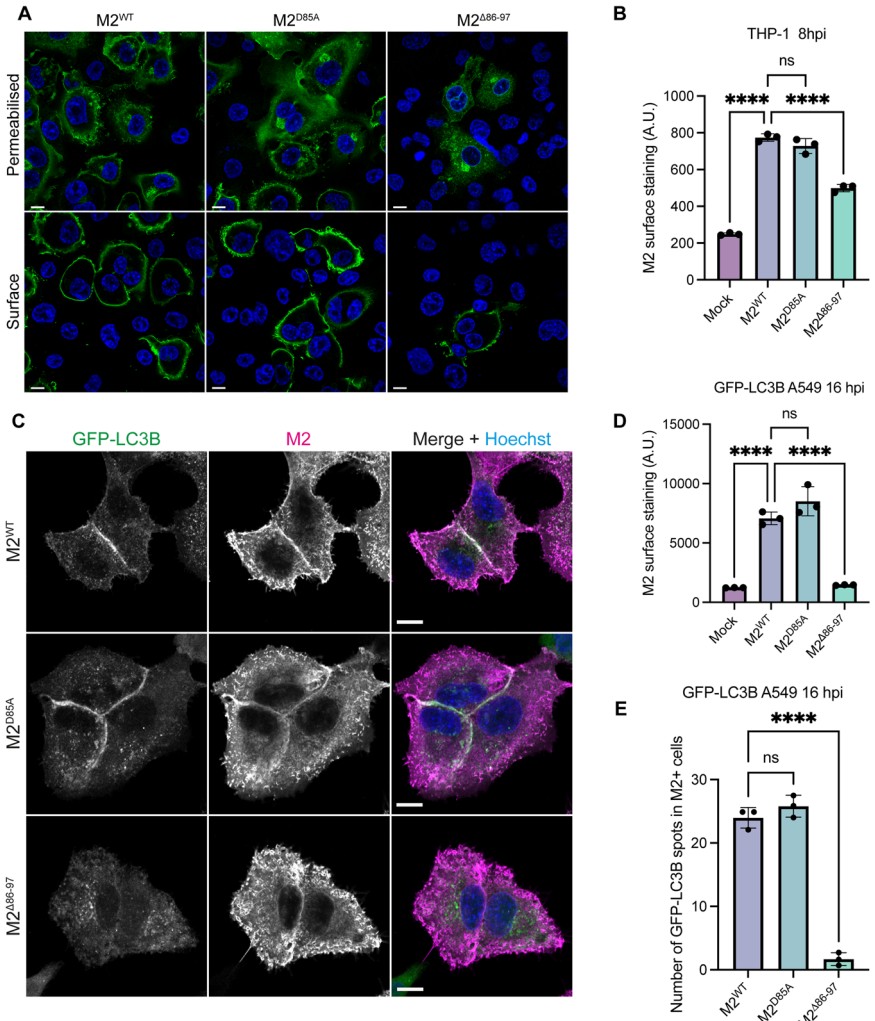

**Figure 3. Cleaved M2 is impaired in plasma membrane relocalisation.**

(A) Representative images of THP-1 cells infected IAV PR8 M2^WT, M2^D85A, and M2^Δ86-97 mutants for 8 h with an MOI of 1. Samples were stained with M2 (in green) and Hoechst (blue). Scale bar represents 10 μm. (B) Quantification of M2 intensity in surface-stained THP-1 cells. Samples were infected with IAV PR8 M2^WT, M2^D85A, and M2^Δ86-97 for 8 h with an MOI of 1. Cells were fixed but not permeabilized, and stained with an antibody against M2. Bars show mean ± SD of $n = 3$ technical replicates. ****M2^WT vs. Mock $P < 0.0001$ and M2^WT vs. M2^Δ86-97 $P < 0.0001$. Ordinary one-way ANOVA with Dunnett's multiple comparisons. (C) Representative images of GFP-LC3B A549 cells infected with IAV PR8 M2^WT, M2^D85A, and M2^Δ86-97 mutants for 16 h with an MOI of 10. Images show GFP-LC3B (green), M2 (magenta), and Hoechst (blue). Scale bar represents 10 μm. (D) Quantification of M2 intensity in surface-stained GFP-LC3B A549 cells. Samples were infected with IAV PR8 M2^WT, M2^D85A, and M2^Δ86-97 for 16 h with an MOI of 10. Bars show mean ± SD of $n = 3$ technical replicates. ****M2^WT vs. Mock $P < 0.0001$ and M2^WT vs. M2^Δ86-97 $P < 0.0001$. Ordinary one-way ANOVA with Dunnett's multiple comparisons. (E) Quantification of GFP-LC3B spot number in M2-positive GFP-LC3B A549 cells. Samples were infected with IAV PR8 M2^WT, M2^D85A, and M2^Δ86-97 for 16 h with an MOI of 10. Bars show mean ± SD of $n = 3$ technical replicates. ****$P < 0.0001$. Ordinary one-way ANOVA with Dunnett's multiple comparisons. Source data are available online for this figure.

affected by cleavage, a plaque assay was performed using supernatant from MDCK cells infected with IAV M2^WT, M2^D85A, and M2^Δ86-97 for 8, 24, and 48 h. No substantial difference in titer between M2^WT, M2^D85A, and M2^Δ86-97 was observed at 8 hpi. While M2^WT and M2^D85A titer remained comparable at later time points, the M2^Δ86-97 mutant showed reduced titer at 24 and 48 hpi (Fig. 5D). Purified M2^D85A and M2^Δ86-97 infectious particles were also used to perform a plaque assay and quantify titer (Fig. 5E). Under these circumstances, the difference in titer between M2^D85A and M2^Δ86-97 mutants, although statistically significant, was less than a log₁₀ fold. This is in contrast to more substantial differences when assaying unpurified supernatant (Figs. 5A,E and EV4D). Taken together,

these results suggest IAV M2^Δ86-97 enters cells and replicates its genome efficiently, but has a defect in assembly or budding.

## IAV M2^Δ86-97 mutant is defective in filamentous budding

IAV is a pleiomorphic virus, forming both filamentous and non-filamentous virions (Badham and Rossman, 2016). While PR8 produces mostly spherical virions (Choppin, 1963; Hayase et al, 1995), a hybrid IAV (MUd) containing genomic segments 1–6 and 8 from PR8 and segment 7 (encoding M1 and M2) from the filamentous A/Udorn/72 strain exhibits a filamentous phenotype (Noton et al, 2007). To determine whether caspase cleavage of M2 acts as a mechanism to

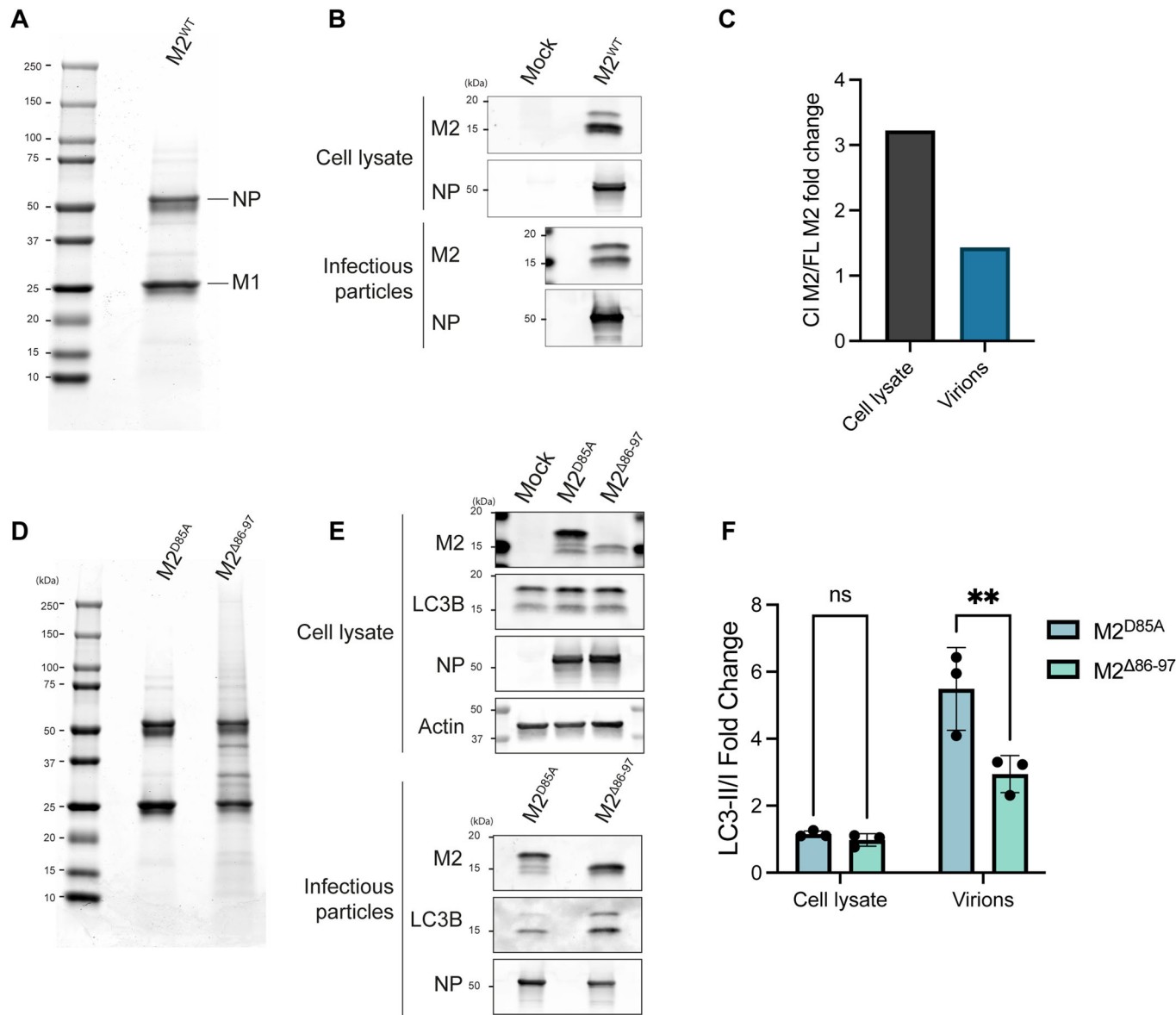

**Figure 4. M2 cleavage reduces M2 incorporation into virions.**

(A) Coomassie-stained gel of purified infectious particles from the supernatant of MDCK cells infected for 48 h with IAV PR8 M2^WT. (B) Representative immunoblots of cell lysate from MDCKs and infectious particles following purification. Cells were infected with IAV PR8 M2^WT for 48 h. The supernatant was then collected and purified. (C) Quantification of ratio of cleaved and full-length M2 in MDCK cell lysate and purified virions, shown in (B). (D) Coomassie-stained gel of purified infectious particles from supernatant of MDCK cells infected for 48 h with IAV PR8 M2^D85A, and M2^Δ86-97 mutant as indicated. (E) Representative immunoblots of cell lysate from MDCKs and infectious particles following purification. Cells were infected with IAV PR8 M2^D85A or M2^Δ86-97 for 48 h. The supernatant was then collected and purified. (F) Quantification of (E). Bars show mean ± SD of $n = 3$ biological replicates. **$P = 0.0038$. Two-way ANOVA with Šidák multiple comparisons test. Source data are available online for this figure.

regulate filamentous and spherical budding, caspase cleavage mutants were generated in a MUd background. While M2^MUd_D85A was rescued, no virus could be produced with permanently cleaved M2^MUd_Δ86-97 using standard reverse genetic techniques. We therefore employed a complementation approach to rescue the M2^MUd_Δ86-97 mutant. HEK293T and MDCK cells were generated to express Udorn M2 under the control of a doxycycline-inducible promoter (Fig. EV5A). M2^MUd_WT(M2_MDCK) and M2^MUd_Δ86-97(M2_MDCK) were successfully rescued using this method, allowing a valid comparison of a truncated M2 in a

filamentous background to its wild-type counterpart (Fig. EV5B). MDCK cells were infected with M2^MUd_WT, M2^D85A and M2^MUd_WT(M2_MDCK), and M2^MUd_Δ86-97(M2_MDCK). M2^MUd_WT and M2^MUd_WT(M2_MDCK) expressed similar levels of M2, and were cleaved to similar extents. Furthermore, M2^MUd_D85A was deficient in M2 cleavage, as previously reported for a PR8 virus, and M2^MUd_Δ86-97(M2_MDCK) only expressed cleaved M2 (Fig. EV5C). To determine whether the presence or absence of the LIR following caspase cleavage affected filament formation, MDCK cells were infected with M2^MUd_WT, M2^D85A,

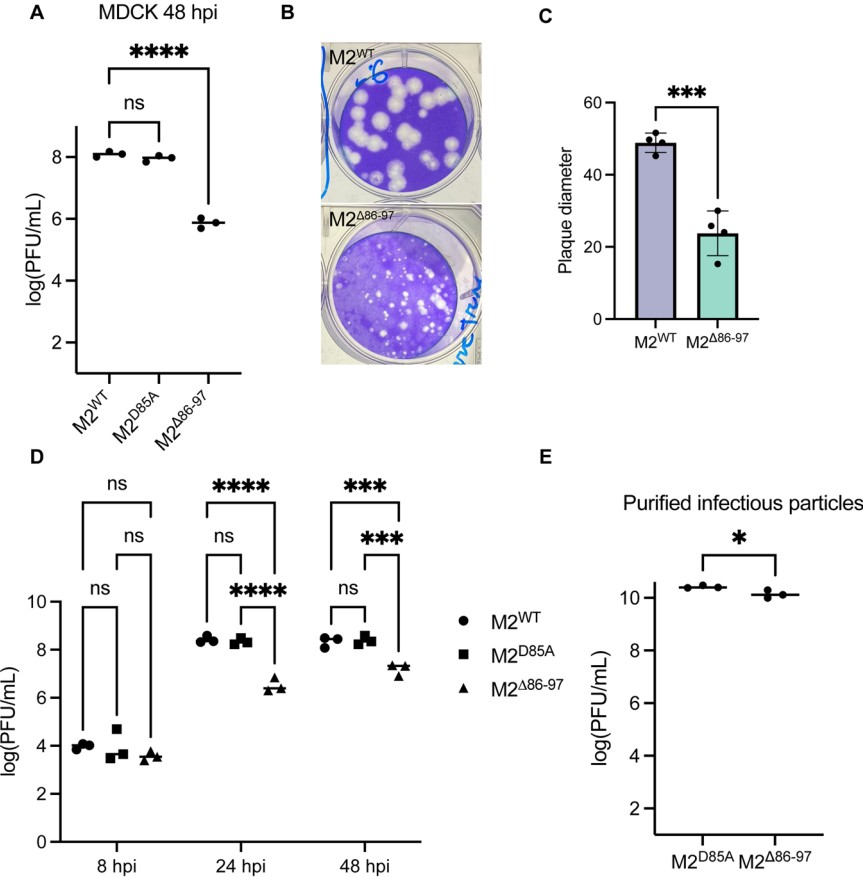

**Figure 5. IAV M2^Δ86-97 is attenuated.**

(A) Plaque assay quantification to assess IAV titer. The supernatant of MDCKs infected with IAV PR8 M2^WT, M2^D85A, and M2^Δ86-97 mutants was collected after 48 h. Plaque assays were performed for 48 h. Bars show mean ± SD of $n = 3$ technical replicates. ****$P < 0.0001$. Log-transformed data are plotted. Ordinary one-way ANOVA with Dunnett's multiple comparisons. (B) Images of plaque assay plates stained with toluidine blue. (C) Quantification of plaque diameter from (H). Bars show mean ± SD of $n = 4$ technical replicates. ***$P = 0.0003$. Unpaired $t$ test. (D) Plaque assay quantification to assess IAV growth kinetics. The supernatant of MDCKs infected with IAV PR8 M2^WT, M2^D85A, and M2^Δ86-97 mutants was collected after 8, 24, and 48 h. Plaque assays were performed for 48 h. Bars show mean ± SD of $n = 3$ technical replicates. ****M2^WT vs. M2^Δ86-97 $P < 0.0001$ and M2^D85A vs. M2^Δ86-97 $P < 0.0001$, ***M2^WT vs. M2^Δ86-97 $P = 0.0004$ and M2^D85A vs. M2^Δ86-97 $P = 0.0002$. Log-transformed data are plotted. Two-way ANOVA. (E) Plaque assay quantification to assess IAV infectious particle plaque forming units/mg of purified protein to assess titer. Purified infectious particles of M2^D85A or M2^Δ86-97 mutants were used. Plaque assays were performed for 48 h. Bars show mean ± SD of $n = 3$ technical replicates. *$P = 0.0338$. The graph shows data as Y=log(Y). Unpaired $t$ test. Source data are available online for this figure.

M2^MUd_WT(M2_MDCK), and M2^MUd_Δ86-97(M2_MDCK) and stained for the expression of plasma membrane HA to visualize filament bundles by light microscopy. While M2^MUd_WT, M2^D85A and M2^MUd_WT(M2_MDCK) formed easily detectable filament bundles, infection with M2^MUd_Δ86-97(M2_MDCK) failed to produce comparable structures (Fig. 6A).

Since standard light microscopy techniques lack sufficient resolution to detect individual filaments, MDCK cells were infected and processed for scanning electron microscopy. Infection with M2^MUd_WT(M2_MDCK) and M2^D85A produced abundant filaments that aggregated in thick bundles made up of individual filamentous virions, often extending to neighboring cells (Fig. 6B). Infection with M2^MUd_Δ86-97(M2_MDCK) produced a strikingly lower number of filaments, consistent with the lack of filament bundle detection by light microscopy (Fig. 6B). These data confirmed the role of the last 11 amino acids of the M2 C-terminal tail for both spherical virion production and filament formation, which suggests a role for caspase cleavage in the regulation of IAV morphology.

## Discussion

Here, we have defined a caspase cleavage site in IAV M2 by mutagenesis ($_{82}$SAVD$_{85}$). Additionally, we have solved a structure of the immediately adjacent LIR motif in the otherwise unstructured IAV M2 C-terminal in complex with LC3B. Mutagenesis of D87 and D88 acidic residues confirmed their contribution to the LIR motif, while having no effect on caspase cleavage. Therefore, the concatenation of caspase cleavage site and LIR suggests that two consecutive protein motifs act together to alter M2 function. This parallels a previous description of caspase-mediated cleavage of the autophagy receptor p62, directly upstream of a LIR motif (Sanchez-Garrido et al, 2018). We hypothesize that this highly conserved interaction and cleavage may act as a regulatory mechanism exploited by IAV to fine-tune virion production in different cellular contexts.

Caspases require cleavage motifs to be accessible, being frequently observed in unstructured regions of proteins (Timmer et al, 2009; Mahrus et al, 2008). This is consistent with the flexible

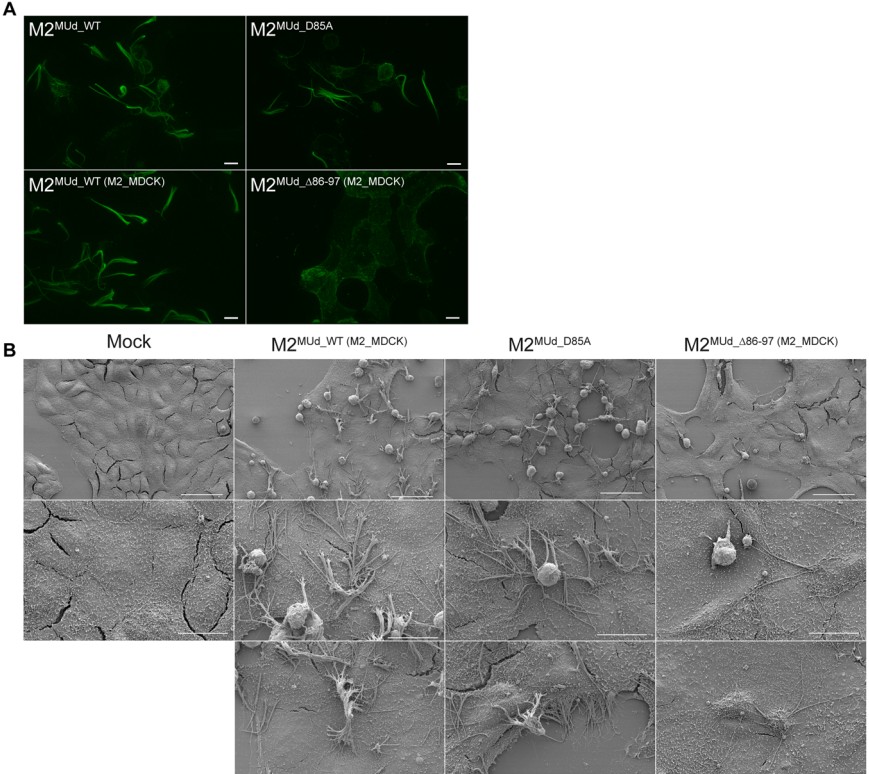

**Figure 6.  IAV M2^Δ86-97 mutant is defective in filamentous budding.**

(**A**) Representative images of MDCK cells infected IAV MUd M2^MUd-WT, M2 ^MUd-D85A, MUd M2 ^MUd-WT(M2_MDCK) (grown in HEK293T and MDCK cells expressing M2 under a doxycycline-inducible promoter) and M2 ^MUd-Δ86-97(M2_MDCK) (grown in HEK293T and MDCK cells expressing M2 under a doxycycline-inducible promoter) mutants for 24 h. Samples were surface-stained with HA (in green). Images were acquired in a z-stack. A maximum projection is shown. Scale bar represents 10 μm. (**B**) Scanning electron microscopy images of MDCK cells. Cells were infected with M2^MUd-WT (M2-MDCK), M2^MUd-D85A, or M2^MUd-Δ86-97(M2-MDCK) for 16 h. Scale bars in the first row of images represent 50 μm, and 20 μm in the remaining images. Source data are available online for this figure.

nature of the caspase cleavage motif observed in the structure of the M2–LC3B interactions. Using CRISPR generated caspase deficient cell lines and chemical inhibition, we show that caspase-6 is largely responsible for M2 cleavage. A partial reduction in M2 cleavage was observed with caspase-3 deficiency and inhibition. This effect may be indirect, since caspase-3 has been reported to cleave caspase-6 for activation (Dagbay and Hardy, 2017). The possibility also remains that while executioner caspases perform distinct roles during apoptosis, they may exhibit a degree of redundancy when it comes to viral substrates (Slee et al, 2001).

Plasma membrane localization of LC3 has previously been reported during IAV infection. This relocalisation is abolished during infection with LIR deficient virus (Beale et al, 2014). Here, we show that M2 and LC3 transport to the plasma membrane can be regulated by caspase-mediated removal of the LIR motif, contextualizing previous findings. IAV uses the host plasma membrane to form its own envelope (Nayak et al, 2009). The C-terminal region of M2 is crucial for viral assembly and budding (Iwatsuki-Horimoto et al, 2006; Schmidt et al, 2013). Here, we show for the first time inclusion of LC3 into IAV virions through interaction with M2. The inclusion of LC3 in virions has been reported in other viruses, although the functional importance for this is unclear (Pena-Francesch et al, 2023; Arnoldi et al, 2014; Taisne et al, 2019; Münz, 2017; Nowag et al, 2014).

The high conservation level of caspase cleavage and LIR motifs highlights the complexity of host machinery regulation by IAV. Caspase cleavage of M2 must provide an evolutionary advantage for the virus despite attenuation observed in the permanently truncated M2^Δ86-97 mutant. While we did not observe an increase in titer with the M2^D85A mutant relative to M2^WT infection, this may reflect a non-rate-limiting abundance of full-length M2 in the cell culture models of infection used in this study. Interestingly, caspase-6 deficient mice exhibited increased susceptibility to IAV infection, with increased mortality and cellular spread (Zheng et al, 2020), suggesting the uncleavable M2^D85A mutant might have increased pathogenicity compared to M2^WT in vivo.

Pathogenic isolates of IAV are pleiomorphic, exhibiting both filamentous and spherical virions (Badham and Rossman, 2016). While the exact role of viral filament formation is unknown, it may aid infection of neighboring cells in the upper respiratory tract (Chu et al, 1949; Badham and Rossman, 2016). Filamentous budding appears to be inefficient, using vastly more host plasma membrane than spherical virions (Badham and Rossman, 2016). It has been previously demonstrated that the M2 LIR motif is required for optimal filamentous budding (Beale et al, 2014). While many of the roles carried out by M2 are associated with its ion channel activity, this is not required for filament formation (Rossman et al, 2010). This is consistent with previous reports

indicating distal C-terminal tail truncations have no significant impact on M2 ion channel activity (Cady et al, 2009; Schnell and Chou, 2008; Nguyen et al, 2008; Tobler et al, 1999). Here, we show that permanently truncated M2, not expressing the LIR motif, is deficient in filament formation. Therefore, we speculate that caspase cleavage of M2 removes the LIR motif to enable a switch between filamentous and non-filamentous budding in response to the depletion of cellular resources. This would provide an evolutionary advantage to the virus, maximizing virion production at later time points.

# Methods

### Reagents and tools table

| Reagent/resource | Reference or source | Identifier or catalog number |
| --- | --- | --- |
| **Experimental models** | | |
| THP-1 cells (*H. sapiens*) | Cell Services STP Francis Crick | N/A |
| HAP1 cells (*H. sapiens*) | Horizon Discovery | HZGHC001508c004 |
| HAP1 CASP3 KO cells (*H. sapiens*) | Horizon Discovery | HZGHC001508c004 |
| HAP1 CASP6 KO cells (*H. sapiens*) | Horizon Discovery | HZGHC001509c004 |
| HAP1 CASP7 KO cells (*H. sapiens*) | Horizon Discovery | HZGHC001510c003 |
| HAP1 CASP8 KO cells (*H. sapiens*) | Horizon Discovery | HZGHC001511c007 |
| HEK293T cells (*H. sapiens*) | ECACC | Cat#12022001; RRID:CVCL_0063 |
| M2-HEK293T cells (*H. sapiens*) | This study | N/A |
| A549 cells (*H. sapiens*) | NCI-DTP | Cat#A549; RRID:CVCL_0023 |
| MDCK cells (*C. familiaris*) | Cell Services STP Francis Crick | N/A |
| M2-MDCK cells (*C. familiaris*) | This study | N/A |
| **Recombinant DNA** | | |
| CASP6_OHu19575_pcDNA3.1(+)-N-DYK | GenScript | NM_001226.4 |
| pENTR-Flag-CASP6 | This work | N/A |
| psPAX2 | Addgene | Cat#12260; Addgene_12260 |
| pMD2-G | Addgene | Cat#12259; Addgene_12259 |
| pLenti-PGK-Flag-CASP6-HygR | This work | N/A |
| pInd20-Ud-M2 | Ulferts et al, 2021 | N/A |
| pDual | Ron Fouchier, Erasmus university Rotterdam NL | N/A |
| pDual-PR8-seg1 | Ron Fouchier, Erasmus university Rotterdam NL | N/A |
| pDual-PR8-seg2 | Ron Fouchier, Erasmus university Rotterdam NL | N/A |
| pDual-PR8-seg3 | Ron Fouchier, Erasmus university Rotterdam NL | N/A |
| pDual-PR8-seg4 | Ron Fouchier, Erasmus university Rotterdam NL | N/A |
| pDual-PR8-seg5 | Ron Fouchier, Erasmus university Rotterdam NL | N/A |
| pDual-PR8-seg6 | Ron Fouchier, Erasmus university Rotterdam NL | N/A |
| pDual-PR8-seg7 | Ron Fouchier, Erasmus university Rotterdam NL | N/A |
| pDual-PR8-seg8 | Ron Fouchier, Erasmus university Rotterdam NL | N/A |
| pDual-Ud-segment 7 | Paul Digard, Roslin Institute | N/A |
| pDual-PR8-seg7_D85A | This work | N/A |
| pDual-PR8-seg7_D87A | This work | N/A |
| pDual-PR8-seg7_D88A | This work | N/A |
| pDual-PR8-seg7_S82A | This work | N/A |

| Reagent/resource | Reference or source | Identifier or catalog number |
| --- | --- | --- |
| pDual-PR8-seg7_V84A | This work | N/A |
| pDual-PR8-seg7_Δ86-97 | This work | N/A |
| pDual-PR8-seg7_FVSI_AAAA | This work | N/A |
| pDual-PR8-seg7_EL_AA | This work | N/A |
| pDual-Ud-segment 7_D85A | This work | N/A |
| pDual-Ud-segment 7_Δ86-97 | This work | N/A |
| **Antibodies** | | |
| Mouse monoclonal anti-IAV Matrix 2 | Abcam | ab5416 |
| Rabbit monoclonal anti-LC3B | Novus Biologicals | NBP2-46892 |
| Rabbit polyclonal caspase-6 | Cell Signaling | 9762 |
| Mouse monoclonal anti-β-actin | Proteintech | 66009-1 |
| Rabbit polyclonal anti-IAV NP | GeneTex | GTX125989 |
| Mouse monoclonal IAV Hemagglutinin (HA) 6F6 | Cambridge Enterprises/ Department of Virology | Amorim et al, 2013; Wright et al, 2009 |
| Rabbit polyclonal PARP-1 | Cell Signaling | 9542 |
| IRDye 680RD Goat anti-mouse | Li-COR Biosciences | 926-68070 |
| IRDye 680RD Goat anti-rabbit | Li-COR Biosciences | 926-68071 |
| IRDye 800CW Goat anti-mouse | Li-COR Biosciences | 926-32210 |
| IRDye 800CW Goat anti-rabbit | Li-COR Biosciences | 926-32211 |
| Donkey anti-goat AF647 | Thermo | A-21447 |
| Goat anti-mouse AF488 | Thermo | A-11001 |
| Donkey anti-mouse AF488 | Thermo | A-21202 |
| Donkey anti-rabbit AF568 | Thermo | A10042 |
| Goat anti-rabbit AF568 | Thermo | A-11011 |
| Goat anti-mouse AF647 | Thermo | A32728 |
| **Oligonucleotides and other sequence-based reagents** | | |
| D85A substitution in PR8 segment 7 | This study | Fwd: AGTGCTGTGGcTGC TGACGAT/Rv: CTGCTGT TCCTTTCGATATTC |
| D87A substitution in PR8 segment 7 | This study | Fwd: GTGGATGCTGcCCG ATGGTCATT/Rv: AGCA CTCTGCTGTTCCT |
| D88A substitution in PR8 segment 7 | This study | Fwd: GATGCTGACGc TGGTCATTTTG/Rv: C ACAGCACTCTGCTGTTC |
| S82A substitution in PR8 segment 7 | This study | Fwd: GGAACAGCA GgctGCTGTGGATG/Rv: TTTCGATATTCTTCCCTC |
| V84A substitution in PR8 segment 7 | This study | Fwd: GCAGAGTGCTgcg GATGCTGACG/Rv: TGTT CCTTTCGATATTCTTCCCTC |
| D86StopD87Stop substitution in PR8 segment 7 | This study | Fwd: TGCTGTGGATtaat aaGATGGTCATTTGTCA GCATAG/ Rv: CTCTGCTG TTCCTTTCGATATTC |
| D85A substitution in UDorn segment 7 | This study | Fwd: GAGTGCTGTGgcgG CTGACGACA/Rv: TGCTGT TCCTTTCGATATTC |
| D86StopD87Stop substitution in UDorn segment 7 | This study | Fwd: TGCTGTGGATtaata aGACAGTCATTTTGTCAG C/Rv: CTCTGCTGTTCCTTTC |
| FVSI90-94AAAA substitution in PR8 segment 7 | This study | Fwd: ggcgTAAAAAACTACC TTGTTTCTAC/Rv: gccgcTA TGCTGACAAAATGACC |
| EL95-96AA substitution in PR8 segment 7 | This study | Fwd: gccgcaGAGCTGGAGT AAAAAACTAC/Rv: agcagcAT GACCATCGTCAGCATC |
| **Chemicals, enzymes, and other reagents** | | |
| Z-VAD-FMK | Invivogen | tlrl-vad |
| Z-VEID-FMK | R&D Systems | FMK006 |
| Z-IETD-FMK | Invivogen | inh-ietd |
| Z-DEVD-FMK | R&D Systems | FMK004 |
| Ampicillin Sodium | Sigma-Aldrich | A9518 |

| Reagent/resource | Reference or source | Identifier or catalog number |
|---|---|---|
| Hygromycin B | Thermo Scientific | 10687010 |
| Doxycycline hyclate | Sigma-Aldrich | D9891 |
| TALON Metal Affinity Resin | Clontech | |
| Thin-wall ultracentrifuge tubes | Beckman Coulter | 331372 |
| Thickwall ultracentrifuge tubes | Beckman Coulter | 355642 |
| 12.7 mm diameter aluminum SEM pin stubs | Labtech | LT-STUB12-100 |
| 96-well CellCarrier plates | PerkinElmer | 6005430 |
| Mini-PROTEAN®TGX | Bio-Rad | 4561091 |
| InstantBlue® Coomassie Protein Stain | Abcam | ab119211 |
| Fetal Calf Serum | GIBCO Life Technologies | A52094 |
| GlutaMax | GIBCO Life Technologies | 35050061 |
| Penicillin–Streptomycin | GIBCO Life Technologies | 15140122 |
| Dulbecco's modified Eagle's medium | GIBCO Life Technologies | 11995065 |
| Iscove's modified Dulbecco's medium | GIBCO Life Technologies | 21980065 |
| Roswell Park Memorial Institute 1640 | GIBCO Life Technologies | 11875093 |
| Phorbol myristate acetate | Invivogen | tlrl-pma |
| Ammonium bicarbonate | Sigma-Aldrich | 40867-50G |
| Phosphoric acid | Sigma-Aldrich | 102409210 |
| Triethylammonium bicarbonate (TEAB) | Sigma-Aldrich | 102335083 |
| Sodium dodecyl sulfate (SDS) | Sigma-Aldrich | 1003528300 |
| Formic acid (FA) 99 + % | Thermo Scientific | 28905 |
| Iodoacetamide (IAA) | Thermo Scientific | 122270050 |
| Tris(2-carboxyethyl)phosphine hydrochloride (TCEP) | Thermo Scientific | PG82080 |
| Trypsin protease MS-grade | Thermo Scientific | 90059 |
| LCMS-grade water | Fisher Chemical | W6-4 |
| LCMS-grade acetonitrile | Fisher Chemical | A955-212 |
| LCMS-grade methanol | Fisher Chemical | A456-4 |
| S-Trap micro columns | ProtiFi | C02-micro |
| TALON Metal Affinity Resin | Clontech | NC9499432 |
| HiLoad 16/600 Superdex 75 column | GE Life Sciences | GE28-9893-33 |
| **Software** | | |
| Fiji (ImageJ) version 2.14.0/1.54f | Schindelin et al, 2012 | RRID:SCR_002285 |
| GraphPad Prism 10 Version 10.1.1 | GraphPadSoftwareLLC | RRID:SCR_002798 |
| Harmony 5.0 | PerkinElmer | RRID:SCR_023543 |
| PyMol | SchrödingerLLC | RRID:SCR_000305 |
| ImageStudioLite | LI-COR Biosciences | RRID:SCR_013715 |
| **Other** | | |
| Zeiss Invert 880 | Zeiss | |
| iSIM microscope | Visitech | |
| Odyssey CLx scanner | LI-COR | |
| Quanta FEG scanning electron microscope | ThermoFisher | |
| Opera Phenix plus | PerkinElmer | |

## Antibodies, reagents, and chemicals

Antibodies used in this study were mouse monoclonal anti-IAV Matrix 2 (Abcam, ab5416, WB 1:1000, IF 1:100), rabbit monoclonal anti-LC3B (Novus Biologicals, NBP2-46892, WB 1:1000), rabbit polyclonal caspase-6 (Cell Signaling, 9762, WB 1:1000), mouse monoclonal anti-β-actin (Proteintech, 66009-1, WB 1:10,000), rabbit polyclonal anti-IAV NP (GeneTex, GTX125989, WB 1:10,000), rabbit polyclonal PARP-1 (Cell Signaling, 9542, WB 1:1000), and mouse monoclonal HA antibody (clone 6F6) (Cambridge Enterprises Department of Virology, (Amorim et al, 2013; Wright et al, 2009)).

Chemical reagents used in this study include Z-VAD-FMK (InvivoGen; tlrl-vad), Z-VEID-FMK (R&D systems; FMK006), Z-DEVD-FMK (R&D systems FMK004), and Z-IETD (Invivogen; inh-ietd) at indicated concentrations. Ammonium bicarbonate, phosphoric acid, triethylammonium bicarbonate (TEAB), and sodium dodecyl sulfate (SDS) were purchased from Sigma-Aldrich (St. Louis, MO). Formic acid (FA) 99%, iodoacetamide (IAA), tris(2-carboxyethyl)phosphine hydrochloride (TCEP), trypsin protease MS-grade were purchased from Thermo Scientific Inc. (Rockford, IL, USA). LCMS-grade solvents: water, acetonitrile (ACN), methanol, were purchased from Fisher Chemical. S-Trap micro columns were purchased from ProtiFi (Huntington, NY). Evotips were purchased from Evosep (Odense, Denmark).

## Cell culture

THP-1 cells (provided by the Cell Services STP of the Francis Crick Institute) were grown in Roswell Park Memorial Institute (RPMI; GIBCO Life Technologies) containing 10% Fetal Calf Serum (FCS), GlutaMax (GIBCO Life Technologies) and Penicillin–Streptomycin (GIBCO Life Technologies). For differentiation, THP-1 cells were treated with 100 ng/mL PMA for 48 h. Media was changed and cells were incubated in fresh media for a further 24 h. HEK293T, MDCK, and A549 cells (provided by the Cell Services STP of the Francis Crick Institute) were cultured in Dulbecco's modified Eagle's medium (DMEM; GIBCO Life Technologies) containing 10% FCS, GlutaMax and Penicillin–Streptomycin. HAP1 cells, including Parental, CASP3 KO (ID: HZGHC001508c004), CASP6 KO (ID: HZGHC001509c004), CASP7 KO (ID: HZGHC001510c003), and CASP8 KO (ID: HZGHC001511c007) cells were purchased from Horizon Discovery, and grown in Iscove's modified Dulbecco's medium (IMDM; GIBCO Life Technologies) containing 10% FCS and Penicillin–Streptomycin. All cell lines were grown in incubators at 37 °C in 5% $CO_2$.

## Plasmids

Plasmids for IAV generation through reverse genetics are described in (Wit et al, 2004). pENTR-Flag-CASP6 was produced using the pENTR/D-TOPO Cloning Kit (Invitrogen) with the CASP6_O-Hu19575_pcDNA3.1(+)-N-DYK plasmid. pLenti-PGK-Flag-CASP6-HygR was generated through gateway cloning with LR clonase (Invitrogen), with pENTR-Flag-CASP6 and pLenti-PGK-GWT-HygR.

## Lentivirus generation

Lentivirus (pLenti-CASP6_OHu19575_pcDNA3.1(+)-N-DYK and pInd20-Ud-M2) was produced by co-transfection of psPAX2 and pMD2-G with PEI in HEK293T cells. Cells were transduced through spinoculation with lentivirus and 8 µg/mL polybrene for 1 h at 500×g at room temperature. Transduced cells were selected with 200 µg/mL Hygromycin B and 400 µg/mL G418.

M2-HEK293T and M2-MDCK cells expressing the IAV Udorn M2 protein under a doxycycline-inducible promoter (following

transduction with pInd20-Ud-M2) were treated with 3 µg/mL doxycycline to induce expression.

## SDS-PAGE and western blot

Cells were grown, treated, or infected in 6-, 12-, or 48-well plates. Before lysis, cells were washed twice in PBS. Cells were then lysed in ice-cold NP-40 buffer (0.5% NP-40, 25 mM Tris–HCl (pH 7.5), 100 mM NaCl, 50 mM NaF) with fresh protease inhibitor cocktail (Sigma). Cell lysate was then collected, clarified (16,200 × g, 4 °C, for 30 min), and the sample protein concentration was assessed through a BCA assay (Pierce). Proteins were resolved on Mini-PROTEAN®TGX gels (Bio-Rad). Staining was carried out with InstantBlue® Coomassie Protein Stain (Abcam) or transferred onto nitrocellulose membranes for western blot.

Nitrocellulose membranes were blocked in 5% dry milk powder in 0.1% Tween-20 for 1 h at room temperature. Primary antibodies were incubated at room temperature for 1 h, or overnight at 4 °C. Membranes were washed in 0.1% Tween-20 and then stained with species-specific IRdye 800CW and 680LT coupled secondary antibodies (LI-COR). Membranes were washed and scanned with an Odyssey CLx scanner (LI-COR).

## Immunofluorescence

Cells were grown in coverslips or in clear, flat-bottom 96-well (PerkinElmer, 6055300), and then fixed with a final concentration of 4% formaldehyde in PBS for 20 min. For samples requiring permeabilization, cells were treated with 0.2% Triton-X100 for 5 min, and washed with PBS. All samples were blocked in 3% BSA in PBS for 30 min. Samples were incubated with primary antibodies for 1 h, washed, and incubated with species-specific AlexaFluor 488 and 647 (Thermo), and Hoescht 33342 for 45 min. Coverslips were mounted onto slides using ProLong™ Glass Antifade Mountant (Invitrogen).

Imaging was done with a Zeiss Invert 880 (63X oil-immersion objective) or a Visitech iSIM microscope (100× oil-immersion objective). Images were processed in Fiji (v.2.3.0/1.53 f; Schindelin et al, 2012) and Adobe Photoshop 2023 (v.24.7.0).

## High-content immunofluorescence quantification

For quantifying the M2 protein, the cells were cultured and infected on 96-well CellCarrier plates (PerkinElmer, 6005430). Plates were then scanned using Opera Phenix plus (PerkinElmer) after staining. The images were acquired utilizing a 40x_water_NA1.1 lens with confocal of 8 planes Z-stacks spanning from −1 to 2.5 µM. The process involved the utilization of excitation lasers at wavelengths of 375, 488 nm, coupled with emission filters at 435–480 and 500–550. The quantification of M2 (488) was analyzed using Harmony 5.0.

## Influenza A virus production and infection

Influenza A virus PR8 (strain A/Puerto Rico/8/1934 (H1N1)) was generated through an eight plasmid-based system in HEK293T cells, as previously described (Wit et al, 2004). Stocks were passaged in MDCK cells, in the presence of TPCK-trypsin (Worthington) and 0.14% BSA Fraction V. For infection, cells were

first washed with serum-free media, and then incubated with virus in serum-free media at 37 °C for an hour. Inoculum was then replaced with fully complemented media and incubated at 37 °C for the indicated period.

## Reverse genetics

Influenza A virus PR8 (strain A/Puerto Rico/8/1934) mutants were generated through the above eight plasmid-based system (Wit et al, 2004). Mutant segment 7 plasmids were generated through Q5 Site-Directed Mutagenesis Kit (New England BioLabs).

## Immunoprecipitation

Cells were incubated in lysis buffer (0.5% NP-40, 10 mM Tris–HCl (pH 7.5), 150 mM NaCl, 0.5 mM EDTA) at 4 °C rotating for 30 min, and clarified (16,200 × g, 4 °C, for 30 min). Dilution buffer (10 mM Tris–HCl (pH 7.5), 150 mM NaCl, 0.5 mM EDTA) was added to lysed samples. Dynabeads™ Protein G for Immunoprecipitation (ThermoFisher 10003D) were washed with PBS-Tween, and incubated with relevant antibodies for 30 min at 4 °C. Samples were then incubated with beads for 2 h rotating at 4 °C. Samples were washed five times, resuspended in sample buffer, and boiled for 5 min at 95 °C before running.

## Crystallization, X-ray data collection, and structure determination

M2 LIR (70–97) and LC3B (3–125) fusion protein in pETM60 were expressed as N-terminal Ubiquitin His tag fusion proteins in *Escherichia coli* BL21 (DE3) (Rogov et al, 2012). Cells were grown at 37 °C to an $OD_{600}$ of 0.5, followed by induction with 0.5 mM isopropyl-β-d-thiogalactoside and further incubation at 20 °C for 20 h. Cells were lysed by sonication in lysis buffer (25 mM Tris, 200 mM NaCl, pH 8.5), and the lysate was cleared by centrifugation (15,000 × g, 4 °C, 40 min). The expressed protein was purified by TALON Metal Affinity Resin (Clontech), cleaved by TEV protease, and purified by size exclusion chromatography (HiLoad 16/600 Superdex 75 column, GE Life Sciences) in 25 mM Tris, 200 mM NaCl, pH 8.5. The crystals of LC3B in complex with M2 LIR were obtained using 30% Jeffamine, 0.1 M HEPES, pH 7.0 by the sitting-drop vapor diffusion method at 293 K. Diffraction data were collected at Swiss Lightsource SLS, beam line PXIII and processed with XDS (Kabsch, 2010). The crystal structure was determined by molecular replacement using the LC3B structure (PDB: 2ZJD) as a search model. Manual model building and refinement were done with Coot, CCP4 software suite and Phenix (Emsley et al, 2010; Winn et al, 2011; Adams et al, 2010). The final statistics of refined models are shown in Table EV1, and the atomic coordinates have been deposited in the Protein Data Bank (PDB ID: 8YV6).

## Plaque assays

To quantify viral titers, virus was serially diluted in serum-free media. In total, 200 µL of diluted virus at different dilutions were added to PBS-washed MDCK cells. Cells were incubated at 37 °C for an hour, with shaking every 15 min. Cells were then overlaid with serum-free media, supplemented with 50% Avicel (RC-581, IMCD), TPCK-trypsin (Worthington) and 0.14% BSA Fraction V,

and incubated for 48 h. The supernatant was then removed, and cells were fixed with Neutral Buffered Formalin for 20 min. Staining was performed with toluidine blue for 20 min.

PFU/mL quantifications were carried out according to $\frac{N}{D \times V}$ where N is the number of plaques, D is the dilution corresponding to the well where the plaques were counted, and V is the volume of inoculum added to each well in mL.

## Viral purification

Viral purification was performed according to (Hutchinson and Stegmann, 2018). MDCK supernatants infected with relevant viruses were spun at 4 °C for 10 min at roughly 2000 × g, in a table-top centrifuge. Media was then transferred to thickwall ultracentrifuge tubes (Beckman Coulter, 355642), placed SW28 rotor, and spun for 10 min at 4 °C at 18,000 × g. The supernatant was gently layered on top of 5 mL of chilled 30% sucrose. Virions were pelleted for 90 min at 4 °C at 112,000 × g. Supernatants were removed, and the pellet was resuspended in 50 μL of 4 °C PBS, and pooled. Resuspended pellets were layered on top of a sucrose density gradient (30-60%) prepared in thin-wall ultracentrifuge tubes (Beckman Coulter, 331372), which were placed in SW41 rotor buckets, and spun 150 min at 4 °C at 209,000 × g. A milky band was visible at 40% sucrose. The band was collected with a Pasteur pipette and ejected into a tube with 9 mL of 4 °C PBS. Finally, virions were pelleted for 60 min at 4 °C at 154,000 × g. The supernatant was discarded, and the remaining pellet was resuspended in 80 μL of 4 °C PBS.

## Proteomic sample preparation

Protein digestion in the S-Trap microcolumn was performed according to the manufacturer's protocol with some modifications. Briefly, 100 μg protein in lysis buffer (10% SDS in 100 mM TEAB pH 8.5) was reduced by adding TCEP to the protein solution to a final concentration of 5 mM and incubated for 15 min at 55 °C. The alkylation was then performed by adding IAA to the reduced samples to a final concentration of 20 mM and incubated for 30 min at room temperature in the dark. A final concentration of 1.1% (v/v) phosphoric acid followed by sixfold volume of binding buffer (90%, v/v, methanol in 100 mM TEAB) was next added to the protein solution. After vortexing, the solution was loaded into an S-Trap microcolumn. The solution was removed by spinning the column at 2000 × g for 1 min. The column was washed with 400 μL binding buffer three times. Finally, 125 μL of 0.80 mg/mL trypsin in 50 mM ammonium bicarbonate was added to the column and incubated 18 h at 37 °C. Digested peptides were eluted using 80 μL of three buffers consecutively: (1) 50 mM ammonium bicarbonate, (2) 0.2% (v/v) FA in H$_2$O, and (3) 50% (v/v) ACN. The elution solutions were collected in the same tube and dried under a vacuum.

## Proteomic sample acquisition DIA

Dried samples were resuspended in 0.1% (v/v) FA and 200 ng of tryptic peptides were loaded onto Evotip and analyzed using an Evosep One LC (EVOSEP) connected to a timsTOF Pro 2 (Bruker). The Evosep One method was 60 SPD (21-minute gradient, cycle time of 24 min) and the mass spectrometer was operated in DIA-PASEF mode. The DIA-PASEF method consisted of 12 *m/z* windows and a cycle time of 1.37 s. The other mass spectrometer parameters were set as follows: *m/z* range, 262 to 1200, the mobility (1/K0) range was set to 0.60 to 1.60 V.s/cm², and the accumulation and ramp time were 100 ms.

## Proteomic sample search

The viral proteins from IAV strain A/Puerto Rico/8/1934 (Uniprot Taxon ID 211044) and host proteins from *Canis lupus familiaris* (Madin-Darby canine kidney cell line) (Uniprot Taxon ID 9615) FASTA files from the Uniprot database were used to generate a spectral library within DIA-NN 1.8 (Demichev et al, 2020) (Data-Independent Acquisition by Neural Networks) following a library-free search method. Specific search settings included cysteine carbamidomethylation enabled as a fixed modification, N-terminal methionine excision enabled, maximum missed cleavages 1, min precursor + 2, max precursor + 4, neural network classifier was set to single-pass mode, cross run normalization was set to retention time-dependent and library generation was set to smart profiling. All output was filtered at 0.01 false discovery rate (FDR).

## Scanning electron microscopy

Samples were fixed in 8% (v/v) formaldehyde (Taab Laboratory Equipment Ltd) in 0.1 M phosphate buffer (pH 7.4) for 20 min at room temperature before being transferred to 2.5% glutaraldehyde/ 4% (v/v) formaldehyde (Taab Laboratory Equipment Ltd) in 0.1 M phosphate buffer (pH 7.4) for a further 30 min at room temperature. Samples were then washed in 0.1 M phosphate buffer and post-fixed in 2% aqueous (v/v) osmium tetroxide (Taab)/1.5% (v/v) potassium ferricyanide (Sigma) for 60 min on ice and then washed twice in ddH$_2$0 followed by a wash in 25% ethanol (EtOH). Samples were dehydrated stepwise for 5 min each in 50%, 70%, 80% and 90% EtOH before a further 2 ×5 min in 100% EtOH. The samples were critical point dried from 100% EtOH in a Leica EM CPD300 critical point dryer, mounted onto 12.7 mm diameter aluminum SEM pin stubs (Labtech) with a carbon tab (Agar Scientific) and coated in 5 nm platinum using a Quorum Q150 R S sputter coater. Samples were imaged in a Quanta FEG scanning electron microscope (ThermoFisher).

## Statistical analysis

Statistical analysis was performed in GraphPad Prism 10 Version 10.1.1 (270). Statistical tests used, error bars, number of replicates, and statistical significance are reported in figure legends.

# Data availability

The final statistics of refined models are shown in Table EV1, and the atomic coordinates have been deposited in the Protein Data Bank (PDB ID: 8YV6):

The source data of this paper are collected in the following database record: biostudies:S-SCDT-10_1038-S44319-025-00388-7.

# Peer review information

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

## Acknowledgements

The authors thank the following: Lewis Timimi and other past and present members of the Beale and Shenoy labs for helpful comments and suggestions. The authors thank E. Hutchinson for helpful advice on influenza virion purification. The authors thank the Proteomics, High Throughput Screening, Cell Services, Light Microscopy and Electron Microscopy core facilities (Francis Crick Institute) for support, and the Worldwide Influenza Center for providing seasonal IAV strains for testing. CFN, BM, MJ, CD, AW, and RB were supported by The Francis Crick Institute, which receives its core funding from Cancer Research UK (CC2087), the UK Medical Research Council (CC2087) and the Wellcome Trust (CC2087). AS acknowledges funding from the MRC (MR/T00004X/1). For the purpose of Open Access, the author has applied a CC BY public copyright licence to any Author Accepted Manuscript version arising from this submission.

## Author contributions

**Carmen Figueras-Novoa**: Conceptualization; Data curation; Formal analysis; Investigation; Visualization; Methodology; Writing—original draft; Writing—review and editing. **Masato Akutsu**: Supervision; Investigation. **Daichi Murata**: Investigation. **Anne Weston**: Data curation; Investigation; Methodology. **Ming Jiang**: Data curation; Investigation. **Beatriz Montaner**: Resources; Investigation. **Christelle Dubois**: Investigation. **Avinash Shenoy**: Conceptualization; Supervision; Methodology; Project administration; Writing—review and editing. **Rupert Beale**: Conceptualization; Supervision; Funding acquisition; Writing—original draft; Project administration; Writing—review and editing.

Source data underlying figure panels in this paper may have individual authorship assigned. Where available, figure panel/source data authorship is listed in the following database record: biostudies:S-SCDT-10_1038-S44319-025-00388-7.

## Funding

## Disclosure and competing interests statement

The authors declare no competing interests.

# Expanded View Figures

**Figure EV1.  Extended data supporting "M2 is cleaved at SAVD motif by caspases".**

(A) Quantification of Fig. 1C. Bars show mean ± SD of $n = 3$ biological replicates. **$P = 0.0065$. Ordinary one-way ANOVA with Bartlett's multiple comparisons. (B) Representative immunoblots of lysates of wild type (WT), Δ*CASP6*, and Δ*CASP6* stably expressing Caspase-6. Indicated samples were infected for 16 h with IAV PR8 at an MOI of 10. (C) Quantification of (B). Bars show mean ± SD of $n = 3$ biological replicates. **$P = 0.0011$. Ordinary one-way ANOVA with Dunnett's multiple comparisons. (D) Quantification of Fig. 1D. Bars show mean ± SD of $n = 3$ biological replicates. **$P = 0.0064$. Unpaired $t$ test. (E) Representative immunoblots of lysates of THP-1 cells infected with IAV PR8 for 24 h. Indicated samples were treated with DMSO as a control, or 50 µM pan-caspase inhibitor (Z-VAD-FMK), caspase-3 inhibitor (Z-DEVD-FMK), and caspase-8 inhibitor (Z-IETD-FMK) as indicated. (F) Quantification of (E). Bars show mean ± SD of $n = 3$ biological replicates. **$P = 0.0041$. Ordinary one-way ANOVA with Dunnett's multiple comparisons. (G) Multiple sequence alignment of amino acids 82-97 in IAV M2. IAV PR8 M2$^{WT}$, M2$^{D85A}$, and M2$^{Δ86-97}$ sequences were produced with Sanger sequencing and aligned to Influenza A virus (A/Puerto Rico/8/34(H1N1)) segment 7 (Schickli et al, 2001). (H) Quantification of Fig. 1F. Graph shows data points for $n = 2$ biological replicates.

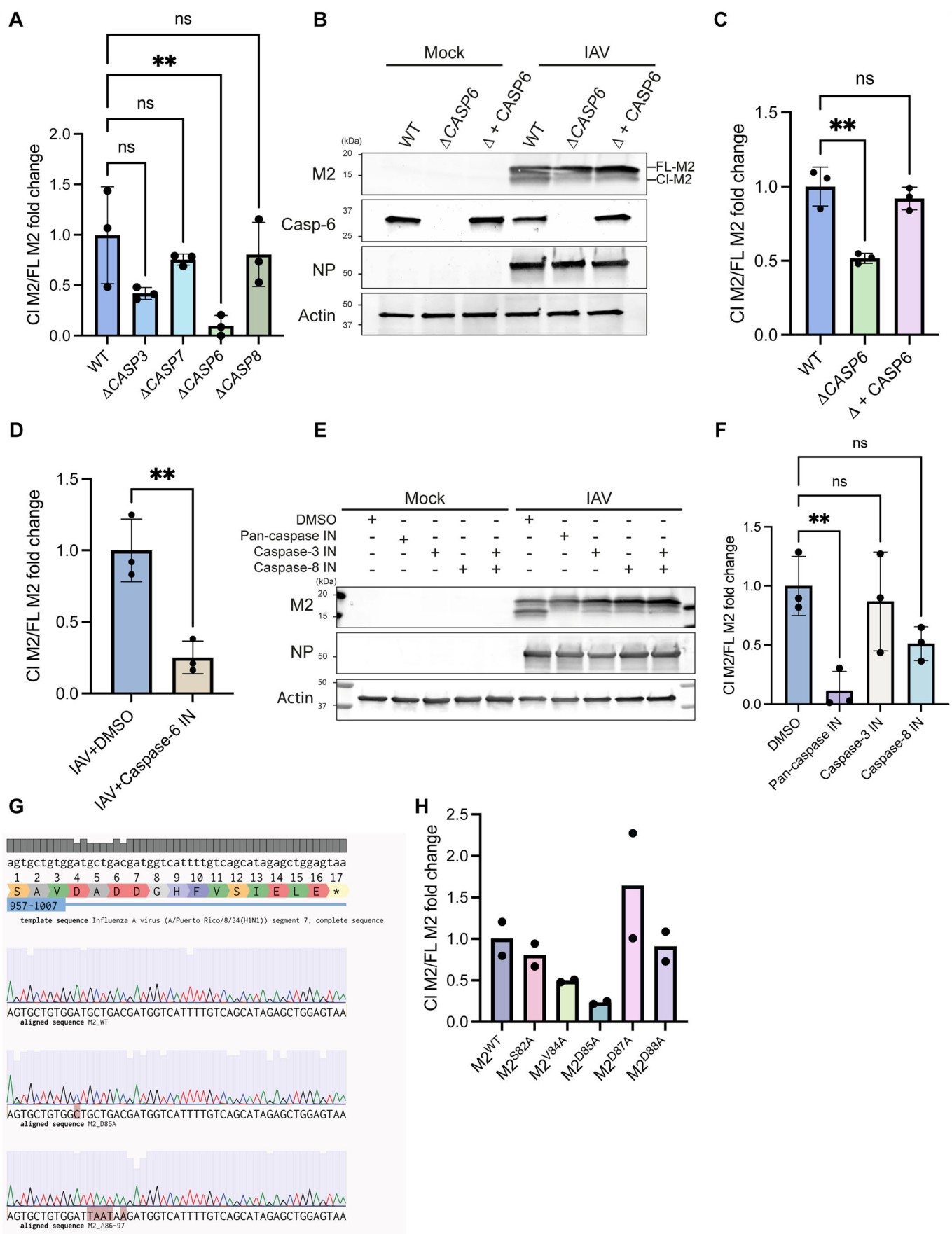

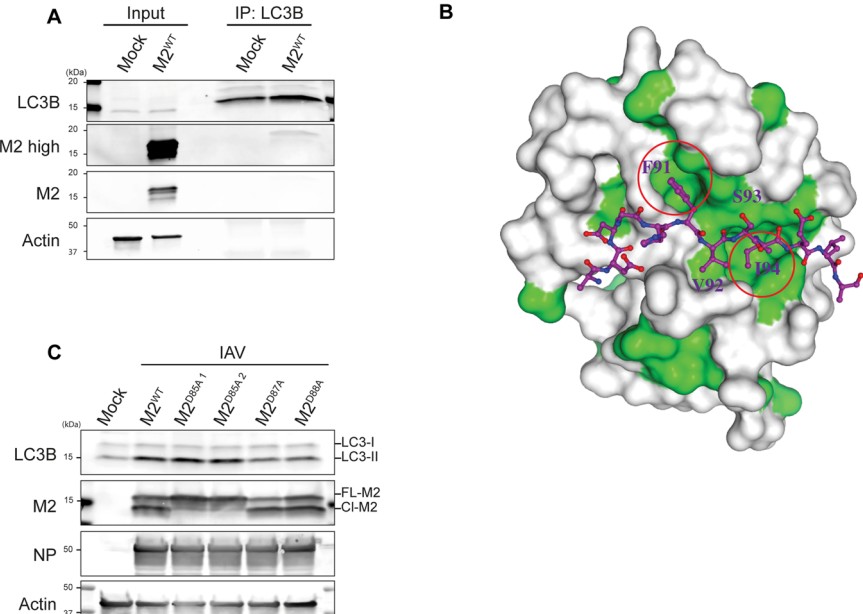

**Figure EV2. Extended data supporting "M2 cleavage disrupts M2–LC3 interaction".**

(A) Immunoprecipitation of endogenous LC3B from A549 cells analyzed by western blotting. Indicated samples were infected for 24 h with IAV PR8 WT, or mutant strains, with an MOI of 10. (B) Surface representation model of M2–LC3B LIR complex. Hydrophobic residues of LC3B are colored green. Red circles indicate hydrophobic pockets of LC3B for LIR interaction. (C) Representative immunoblots of lysates of THP-1 cells infected with IAV M2$^{WT}$, M2$^{D85A}$, M2$^{D87A}$and M2$^{D88A}$ mutants for 24 h with an MOI of 10.

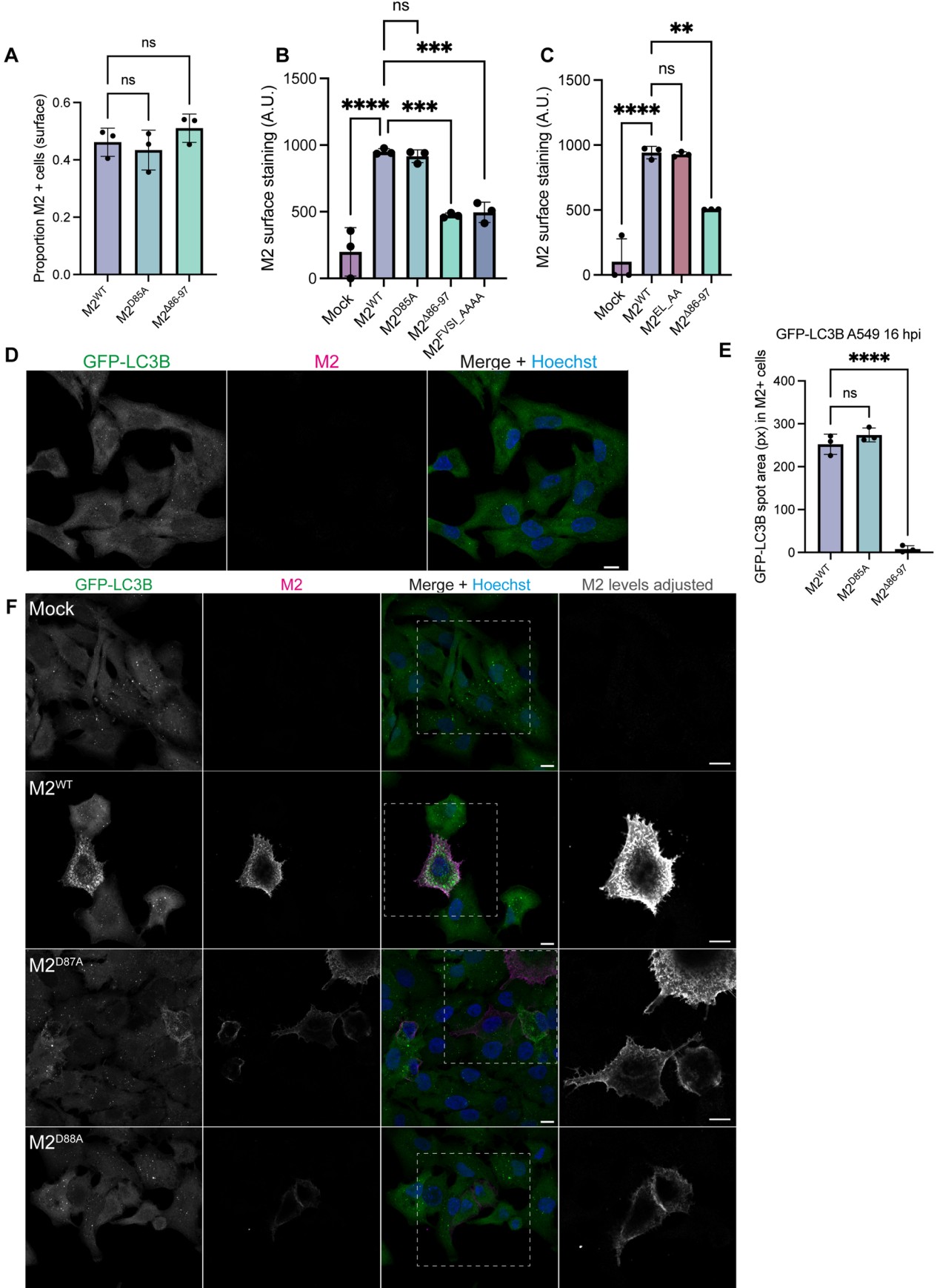

◀ **Figure EV3. Extended data supporting "Cleaved M2 is incorporated into virions at a lower rate and exhibits a decreased titer".**

(A) Quantification of the proportion of M2-positive cells in surface-stained THP-1 cells. Samples were infected with IAV PR8 M2$^{WT}$, M2$^{D85A}$, and M2$^{\Delta86-97}$ for 8 h with an MOI of 1. Bars show mean ± SD of $n = 3$ technical replicates. Ordinary one-way ANOVA with Dunnett's multiple comparisons. (B) Quantification of M2 intensity in surface-stained THP-1 cells. Samples were infected with IAV PR8 M2$^{WT}$, M2$^{D85A}$, M2$^{\Delta86-97}$, and M2$^{FVSL\_AAAA}$ for 8 h with an MOI of 1. Bars show mean ± SD of $n = 3$ technical replicates. ****$P < 0.0001$, ***: M2$^{WT}$ vs. M2$^{\Delta86-97}$ $P = 0.0003$ and M2$^{WT}$ vs. M2$^{FVSL\_AAAA}$ $P = 0.0004$. Ordinary one-way ANOVA with Dunnett's multiple comparisons. (C) Quantification of M2 intensity in surface-stained THP-1 cells. Samples were infected with IAV PR8 M2$^{WT}$, M2$^{EL\_AA}$, and M2$^{\Delta86-97}$ for 8 h with an MOI of 1. Bars show mean ± SD of $n = 3$ technical replicates. ****$P < 0.0001$, **$P = 0.0010$. Ordinary one-way ANOVA with Dunnett's multiple comparisons. (D) Representative images of GFP-LC3B A549 cells mock-infected from the experiment shown in Fig. 3C. Images show GFP-LC3B (green), M2 (magenta), and Hoechst (blue). Scale bar represents 10 μm. (E) Quantification of GFP-LC3B spot area in pixels in M2-positive GFP-LC3B A549 cells. Samples were infected with IAV PR8 M2$^{WT}$, M2$^{D85A}$, and M2$^{\Delta86-97}$ for 16 h with an MOI of 10. Bars show mean ± SD of $n = 3$ technical replicates. ****$P < 0.0001$. Ordinary one-way ANOVA with Dunnett's multiple comparisons. (F) Representative images of surface-stained GFP-LC3B A549 cells that have either been mock-treated or infected with IAV PR8 M2$^{WT}$, M2$^{D87A}$, and M2$^{D88A}$ mutants for 16 h with an MOI of 10. Images show GFP-LC3B (green), M2 (magenta), and Hoechst (blue). Dashed box highlights the area of crop. M2 images were also adjusted to the same levels across infections to illustrate differences in intensity in M2 surface staining. Scale bar represents 10 μm.

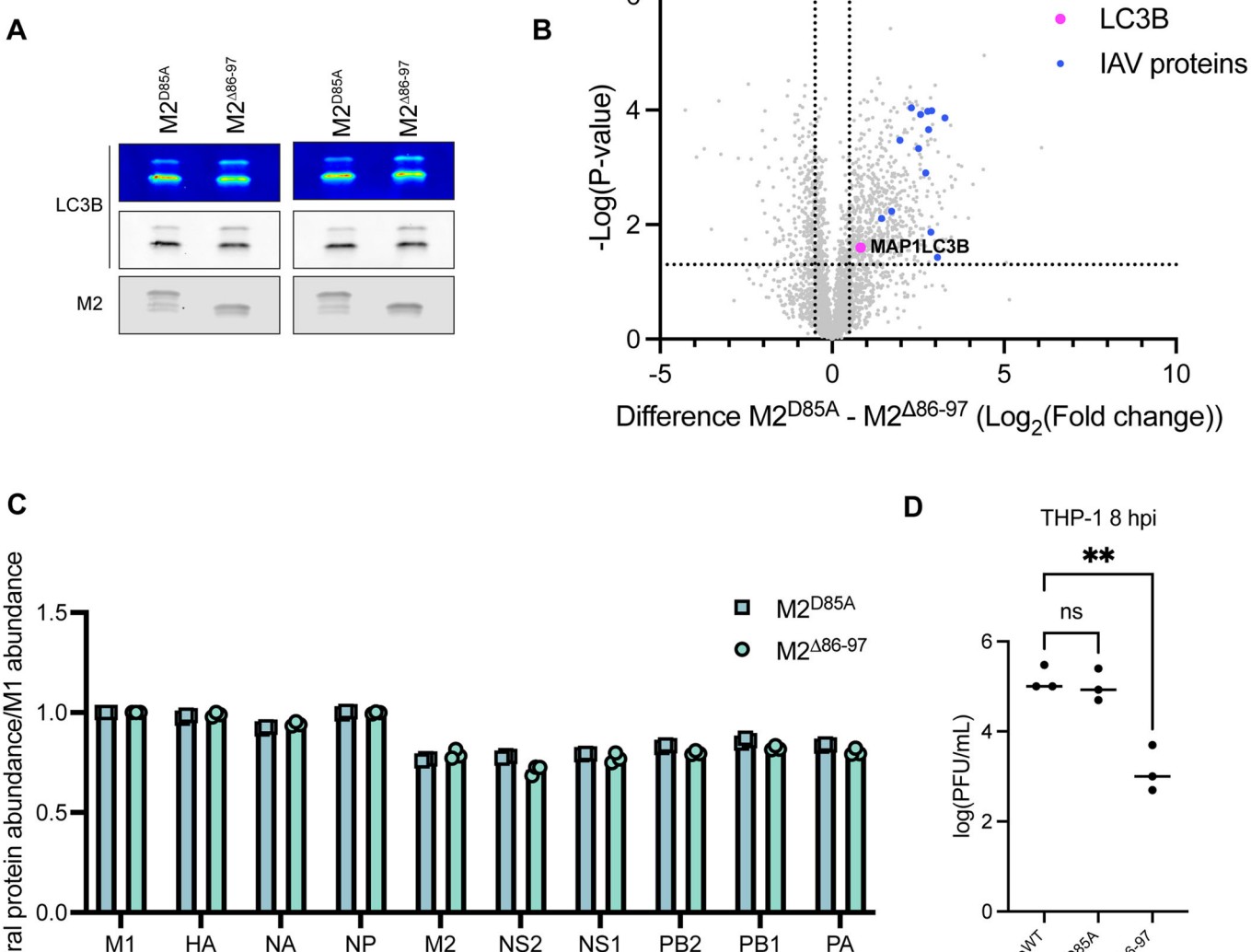

**Figure EV4. Extended data supporting "M2 cleavage reduces M2 incorporation into virions" and "IAV M2$^{Δ86-97}$ is attenuated".**

(A) Biological replicates of immunoblots in Fig. 4E showing lysate of infectious particles following purification. Cells were infected with IAV PR8 M2$^{D85A}$ or M2$^{Δ86-97}$ for 48 h. The supernatant was then collected and purified. (B) Volcano plot showing the difference in log$_2$(fold change) of the abundance of proteins expressed in M2$^{D85A}$ purified virions when comparing them to M2$^{Δ86-97}$ purified virions from 3 biological replicates. MAP1LC3B is shown as a pink dot, and Influenza A proteins are shown as blue dots. Dashed lines represent a difference in log$_2$(fold change) of more than 0.5 or less than -0.5, and -Log(P-value) of 1.3). Unpaired $t$ test. (C) Abundance of viral proteins was calculated through normalization to M1 abundance and compared between M2$^{D85A}$ and M2$^{Δ86-97}$ abundance. Bars show mean ± SD of $n = 3$ biological replicates. (D) Plaque assay quantification to assess IAV titer following THP-1 infection. The supernatant of THP-1 cells infected with IAV PR8 M2$^{WT}$, M2$^{D85A}$, and M2$^{Δ86-97}$ mutants was collected after 8 h. Plaque assays were performed for 48 h. Bars show mean ± SD of $n = 3$ biological replicates. **$P = 0.0013$. Graph shows data as Y=log(Y). Ordinary one-way ANOVA with Dunnett's multiple comparisons.

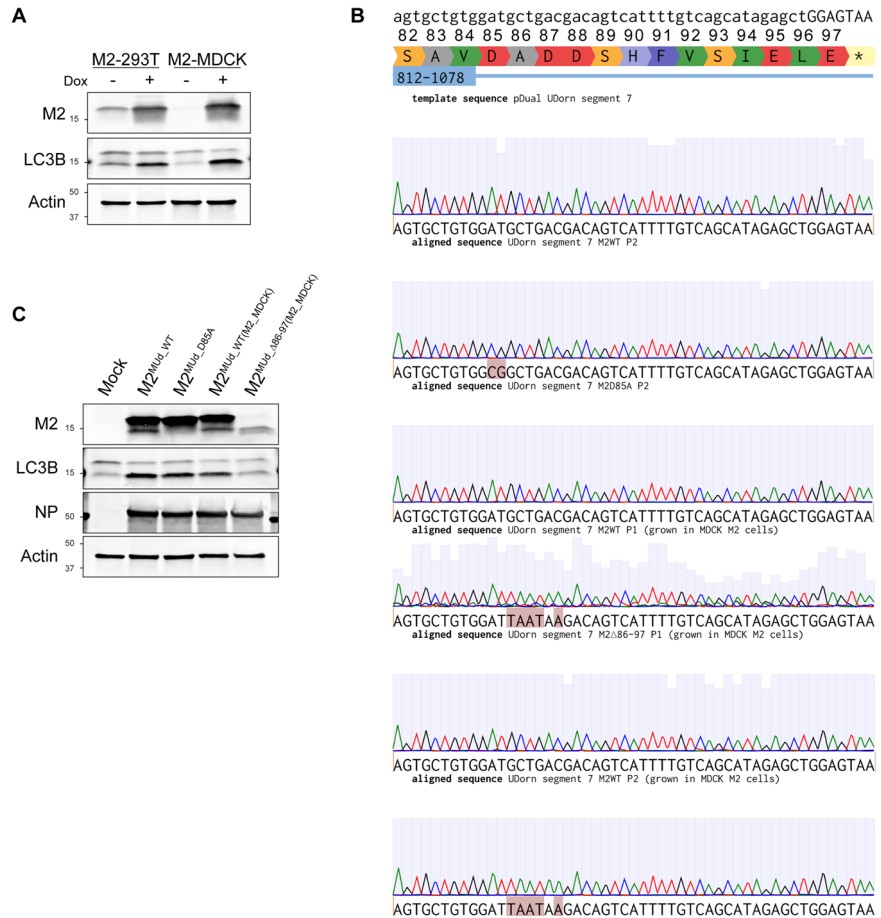

**Figure EV5. Extended data supporting "IAV M2^Δ86-97 mutant in a filamentous background is defective in filamentous budding".**

(A) Immunoblot of lysates of HEK293 and MDCK cells expressing a doxycycline-inducible promoter. Indicated cells were treated with doxycycline for 24 h. (B) Multiple sequence alignment of amino acids 82-97 in IAV M2. IAV MUd M2^MUd-WT, M2^MUd-D85A, M2^MUd-WT(M2_MDCK) (grown in HEK293T and MDCK cells expressing M2 under a doxycycline-inducible promoter), and M2^MUd-Δ86-97(M2_MDCK) (grown in HEK293T and MDCK cells expressing M2 under a doxycycline-inducible promoter). Sequencing of two viral passages is shown for M2^MUd-WT(M2_MDCK) and M2^MUd-Δ86-97(M2_MDCK) to verify amino acid substitutions leading to truncation have not been reversed. Sequences were produced with Sanger sequencing and aligned to pDual A/Udorn/307/1972 H3N2 segment 7. (C) Immunoblot of lysates of MDCK cells infected with IAV MUd M2^MUd-WT, M2^MUd-D85A, M2^MUd-WT(M2_MDCK), and M2^MUd-Δ86-97(M2_MDCK) for 24 h.

