## [Peer Review File · EMBO Reports]

Caspase cleavage of Influenza A virus M2 disrupts M2-LC3 interaction and regulates virion production

Carmen Figueras-Novoa, Masato Akutsu, Daichi Murata, Anne Weston, Ming Jiang, Beatriz Montaner, Christelle Dubois, Avinash Shenoy, and Rupert Beale

Corresponding author(s): Rupert Beale (rupert.beale@crick.ac.uk) , Avinash Shenoy (a.shenoy@imperial.ac.uk)

Review Timeline:

Transfer Date:	2nd Jul 24
Editorial Decision:	4th Jul 24
Revision Received:	30th Oct 24
Editorial Decision:	25th Nov 24
Revision Received:	28th Jan 25
Accepted:	31st Jan 25

Editor: Achim Breiling

Transaction Report: This manuscript was transferred to EMBO reports following peer review at Review Commons.

**Review
COMMONS**

Revision Plan

Manuscript number: RC-2024-02467

Corresponding authors: Rupert Beale, Avinash Shenoy

We thank the reviewers for their positive assessment of our manuscript. We agree that there are some further experiments suggested by the reviewers that would enhance our study. We have highlighted further proposed experimental work in **bold** for clarity.

Reviewer #1 (Evidence, reproducibility and clarity (Required)):

1. EVIDENCE, REPRODUCIBILITY AND CLARITY

Summary:

The Matrix 2 (M2) protein of influenza A virus (IAV) is a single pass transmembrane protein known to act as a tetrameric ion channel that is important for both viral entry and egress. The paper by Figueras-Nova et al. entitled "Caspase cleavage of Influenza A virus M2 disrupts M2-LC3 interaction and regulates virion production" reports on the regulation of IAV virion production through a regulatory interplay between a caspase cleavage site and a LC3 interacting region (LIR) motif in M2. In its C-terminal cytoplasmic tail the IAV M2 protein contains a C-terminal LIR motif interacting with LC3. The authors show that this LIR motif is preceded by a functional caspase cleavage motif cleaved predominantly by caspase-6, with some contribution from caspase-3: The motif 82-SAVD-85 directs cleavage after the aspartate (D) at position 85. The cleavage leads to loss of the remaining C terminal sequence from amino acid 86 to 97. The core LIR motif 91-FVSI-94 LIR motif is then lost from M2 which can no longer bind LC3. As previously described by the same group using point mutations in the LIR motif (Ref 12.), loss of a functional LIR., here by caspase-mediated deletion of the LIR, affects the virion production and inhibits filamentous budding. LC3B lipidation is increased upon treatment with a caspase inhibitor. The authors show for the first time that LC3 is included into IAV virions via binding to M2. Furthermore, they also report a co-crystal structure of the M2 C terminus (aa 70-97), containing the caspase cleavage site and LIR, and LC3B (aa 3-125) adding new insights into this interaction and showing that the caspase cleavage site is in a flexible region N-terminal to the LIR. This work shows how caspase cleavage may modulate LC3B lipidation, trafficking to the plasma membrane, incorporation of LC3B in the virions, filamentous budding and virion production (viral titer).

Major comments:

The findings reported here are very well supported by the data shown. This is a very clearly written paper with well described and nicely visualized results that are accompanied by adequate statistical analyses.

We thank the reviewer for their assessment of our manuscript.

The authors report a new way the LC3B binding to the C-terminal tail of the M2 proteins is regulated

Revision Plan

and suggest that this is an adaptation the virus has made to adjust virion production to host cell status by hijacking the function of host caspases. They show that the caspase cleavage motif is evolutionary conserved and use that as an argument. Perhaps it could be discussed if it also could be an argument that the host protects itself against a too massive virion production as this could be too detrimental to the host? Would it not also be an evolutionary advantage to the virus in the long run by avoiding killing the host?

This is an interesting point. We agree there could be advantage for the virus not to overproduce virions under certain circumstances. Consistent with this caspase-6 deficient mice had increased mortality in response to IAV PR8 infection, and presented and increase in viral spread in the lungs (Zheng, 2021; doi: [10.1016/j.cell.2020.03.040](https://doi.org/10.1016/j.cell.2020.03.040)). This is also relevant for the comments made by Reviewer 2. The manuscript will be updated to include a discussion of this point.

A question I may raise which is optional as it may be too much work to address as part of this study is if the reported regulation of LC3B binding has any role in regulating the ion channel function of the M2 tetramer?

It is well established that there is no impact of distal C-terminal truncations on M2 ion channel activity (Cady et al., 2009, doi: [10.1021/bi9008837](https://doi.org/10.1021/bi9008837) Schnell and Chou, doi: [10.1038/nature06531](https://doi.org/10.1038/nature06531); Nguyen et al., 2008, doi: [10.1021/bi801315m](https://doi.org/10.1021/bi801315m); Tobler et al., 1999, doi: [10.1128/jvi.73.12.9695-9701.1999](https://doi.org/10.1128/jvi.73.12.9695-9701.1999)). This is also consistent with data from our lab (Ulferts et al., 2021, doi: [10.1016/j.celrep.2021.109899](https://doi.org/10.1016/j.celrep.2021.109899), Beale et al., 2014, doi: [10.1016/j.chom.2014.01.006](https://doi.org/10.1016/j.chom.2014.01.006)) as well as others (Ren et al., 2015, doi: [10.1128/JVI.00576-15](https://doi.org/10.1128/JVI.00576-15)) showing the effects of the LIR motif and the proton channel are distinct. We appreciate the reviewer suggesting further work here as optional, but there is already compelling evidence to show there is no substantial effect of the LIR motif on ion channel activity. (See also Reviewer 2 points 4 and 5).

Minor comments:

Delete "with" in line 145.

This will be changed in the updated manuscript.

Line 217: It should be written more specifically how "cells were surface stained with M2"

The protocol for surface staining of M2 will be explained in more detail in the updated manuscript.

In the Introduction a description of what filamentous vs "spherical" budding is, could perhaps be included as I missed that reading through, although it comes in the end of the Discussion.

A description of the pleomorphic nature of IAV will be added to the Introduction.

Revision Plan

Reviewer #1 (Significance (Required)):

2. SIGNIFICANCE

This is a very well performed study with a sound experimental strategy and well performed assays with clear results increasing our insight into the interplay between the Influenza A virus and host cells. Although caspase mediated cleavage of the autophagy receptor and signaling scaffold protein p62 (Ref. 25), removing the LIR and LC3-binding, has been reported before I consider this study as novel in reporting this type of regulation of LC3 binding. The cleavage of p62 deletes a large part of the protein while here it is a "clean" deletion of the LIR sequence representing a conceptual advance of regulation of LC3 binding.

The study also reports for the first time on LC3B incorporated into virions.

The effects on trafficking to the plasma membrane and viral budding and virion production are similar to those reported before (Ref. 12) using viruses with point mutations crippling the LIR motif. This research will be of interested to all studying virus- host interaction and to the autophagy field both as a non autophagic role of LC3B, and as a regulatory mechanism of LIR-LC3B interactions involving the irreversible caspase cleavage-mediated deletion of the LIR motif.

We thank the reviewer for this assessment of our manuscript.

Reviewer #2 (Evidence, reproducibility and clarity (Required)):

The influenza A virus (IAV) M2 protein is small transmembrane protein which plays a role in virus entry and egress. In a previous study, Beale et al. (2014) identified an LC3-interacting region (LIR) in the M2 cytoplasmic domain that was found to recruit the LC3B protein to the plasma membrane. Recombinant IAV harboring mutations in the LIR motif showed reduced particle stability and lost filamentous morphology.

In the present study, Figueras-Novoa et al. show that the LIR motif is removed in response to activation of cellular caspases. The authors demonstrate that in in IAV-infected THP-1 cells M2 is partially cleaved at the motif (82)SAVD(85)↓A by caspase 6. Caspase inhibitors abolished cleavage, and a mutant virus harboring the D85A substitution was found to be resistant to caspase action. A crystal structure of purified M2 C- terminus and LC3B revealed that the caspase cleavage site lies in a flexible region that is accessible to caspases.

Mutant virus encoding a truncated M2 protein (M2 Δ 86-97) was unable to interact with LC3, in accordance with the absence of the LIR motif. The M2 Δ 86-97 mutant showed reduced lipidation of LC3, while enhanced lipidation of LC3 was observed when wild-type virus-infected cells were treated with caspase inhibitors. The authors also observed that cell surface transport of M2 Δ 86-97 but not M2-D85A was impaired. However, in purified virus particles a mix of cleaved and uncleaved

Revision Plan

M2 was detected. The authors also demonstrated that lipidated LC3B was present in purified virions of wild-type virus particles but even more abundant in M2-D85A virions. Finally, M2 Δ 86-97 mutants produced significantly less infectious particles compared to wild-type virus while the D85A cleavage mutant replicated to similar titers than wt virus.

Based on these findings the authors concluded that caspases regulate the interaction of M2 protein with LC3 which impacts virion production. Specifically, they propose that caspase-mediated removal of the LIR motif may enable a switch between filamentous and non-filamentous budding in response to depletion of cellular resources. However, the authors were unable to rescue a filamentous IAV with a truncated M2 protein and therefore could not provide direct proof for their guess.

While the data are sound and presented well, they do not support the conclusions of the authors.

1. To the authors opinion, the conserved caspase cleavage site in the M2 protein might provide an evolutionary advantage for the virus. However, the M2-D85A mutation has no effect on viral replication, so the biological significance of why M2 needs to be cleaved at all is unclear. The conclusion that caspase-induced M2 cleavage is a fine-tuning mechanism of IAV has not been supported by experiments.

We thank the reviewer for the assessment of our data. We think the reviewer is specifically objecting to the phrase “We conclude that this highly conserved interaction and cleavage act as a regulatory mechanism exploited by IAV to fine-tune virion production in different cellular contexts.” This is a reasonable inference from our results, but we accept that it is not proven. We will change the wording to make it clear this has not been definitively demonstrated.

2. The finding that the permanently truncated IAV M2 mutant virus was substantially attenuated does not necessarily mean that abrogation of M2-LC3 interaction was responsible for this attenuation. As the M2 protein plays a role in virus budding at the plasma membrane (recruitment of M1 protein, induction of membrane curvature, membrane scission), the impaired transport of the truncated M2 protein might already explain that the virus was attenuated and that incorporation of the protein into the viral envelope was reduced.

We will confirm this further with additional experiments using LIR mutants. Recapitulating the plasma membrane transport defect of truncated M2 with LIR mutants including the newly characterised M2^{D87A} and M2^{D88A} mutants and a more severe mutant with a FVSI_AAAA substitution would strongly imply this truncation mutant phenotype is due to the lack of LIR motif.

Revision Plan

3. It is also not clear whether the loss of the C-terminal 11 amino acids may have affected the interaction of the M2 protein with other proteins such as TRAPPC6A-delta (Zhu et al., 2017).

This is a reasonable point, however Zhu et al., 2017 (<https://doi.org/10.1128/jvi.01757-16>) reported that the interaction with TRAPPC6A retains M2 intracellularly. If the phenotype observed with our truncation was due to the loss of interaction with TRAPPC6A, the opposite phenotype would be observed (more M2 in the plasma membrane with the truncated M2^{Δ86-97} mutant). **To address this point directly we will attempt to rescue an M2 mutant virus that has disrupted the reported TRAPPC6A binding site and assess M2 plasma membrane localization.**

4. The authors did not rule out whether the truncation of the M2 protein by 11 amino acids would have an effect on proton channel activity. Proton channel activity, however, might be important to preserve the metastable conformation of HA in the secretory pathway and might be also important for virus uncoating.

5. M2^{Δ86-97} induced less LC3 lipidation than wild-type M2 or the D85A mutant. The remaining lipidation was attributed to the ion channel activity of the M2 protein. Can the authors rule out that the truncation of the M2 protein led to reduced ion channel activity which in turn led to reduced LC3B lipidation?

We have addressed points 4 and 5 in response to Reviewer 1.

6. The suggested role of caspase cleavage as a regulatory switch between filamentous and spherical virions (lines 304- 313) is highly speculative as long as the authors do not provide any experimental proof for it. The authors indicated that they were unable to rescue filamentous IAV with M2^{Δ86-97}. However, would it be possible to use caspase inhibitors to test their hypothesis?

We acknowledge that M2^{Δ86-97} could not be rescued in a filamentous background. The use of caspase inhibitors would only increase the amount of full length M2 present, and does not provide an alternative strategy for increasing the proportion of truncated M2. However, since M2^{Δ86-97} mutant could not be rescued, **we will attempt to rescue additional LIR motif mutants to address this point. In particular, D87A and D88A mutants could be generated in a MUD background, as well as the F91S mutant.**

7. The authors used only the PR8 strain for their studies, a highly cell culture-adapted strain with spherical morphology. Are the findings obtained with this strain are also valid for others IAV strains?

As we highlight in Figure 2I, both the caspase cleavage motif and LIR motif are highly conserved in human IAV strains. PR8 was used as it is the reverse genetic system in use and approved for use in the lab. **We will attempt to address this by testing whether other IAV strains we are**

Revision Plan

able to obtain also undergo caspase mediated cleavage of M2. If possible, we will obtain recent clinical isolates to show cleavage of M2 in a strain that has not adapted to cell culture.

8. The authors mainly used the THP-1 cells for their studies, a human macrophage-like cell line. However, human IAV mostly replicate in epithelial cells of the respiratory tract and cause only abortive infections of macrophages. Why did the authors choose this cell line? Can the findings obtained with this cell line be translated to epithelial cells of the airways?

THP-1 cells are widely used for the study of caspase activity. However, we also show M2 cleavage in MDCK cells and HAP1 cells. PR8 infection of A549 cells does not induce significant amounts of cell death in the infection time points used and, as caspase activation is linked to cell death, we did not observe M2 cleavage in this cell type. **We will attempt to infect some epithelial cell types to confirm this phenotype.**

9. Minor issues:

- Fig. 1C: There seem to be quite some differences in the cleavage efficiency of M2 between panels A, B, C, and D? Any explanations?

Different cell types (THP-1 cells and HAP1 cells) are used for the experiments mentioned above, which accounts for the different amount of M2 cleavage.

- Fig. 1: Panel E: The labeling of the first amino acids as aa 76 seems to be wrong!

We thank the reviewer for pointing this out, this will be corrected in the updated manuscript.

- Line 147: ...caspase mediated disruption of the M2-LC3 interaction (Fig 2A-B). Should be Fig. 2A-C.

This sentence was referring to Figure 2A-B, as it refers to LC3B lipidation and not the colIP. This sentence will be changed in the text to reflect the intended meaning.

- Growth kinetics of the various mutant viruses are missing?

We will provide growth kinetics for the relevant mutants (M2^{D85A} and M2^{Δ86-97}).

- Line 195: The authors speculate that aa85 is important for viral fitness: That should be demonstrated!

This speculation is based on the very strong conservation of D85 in human IAV strains. The

Revision Plan

importance of D85 in viral fitness (permitting cleavage of M2) is only likely to be directly demonstrable in transmission models (for example ferrets) which is not feasible or justifiable.

Reviewer #2 (Significance (Required)):

Authors concluded that caspases regulate the interaction of M2 protein with LC3 which impacts virion production. Specifically, they propose that caspase-mediated removal of the LIR motif may enable a switch between filamentous and non-filamentous budding in response to depletion of cellular resources. However, the authors were unable to rescue a filamentous IAV with a truncated M2 protein and therefore could not provide direct proof for their guess.

As stated in the response to the comments above, **we will attempt to rescue LIR mutant viruses (D87A and D88A) in a MUD background which would provide further support for our hypothesis.** Our data has significance for the understanding of the cell biology of influenza infection as commented on by Reviewers 1 and 3.

Reviewer #3 (Evidence, reproducibility and clarity (Required)):

Summary :

In this article, the authors identify a caspase cleavage site in the influenza A virus (IAV) Matrix 2 protein (M2) that leads to a truncated form of M2 deleted from its C-term LC3-interacting region (LIR). This cleaved form of M2 is seen and accumulates starting at 12 hours post-infection. IAV expressing M2 delta 86-97 mutant, corresponding to cleaved M2, seems to disrupt LC3B localization to cell plasma membrane upon infection. The authors also show that the IAV M2 delta 86-97 has a reduced viral titer compared to IAV WT. Overall the data are quite exciting where the authors identify the specific caspase responsible for the cleavage and show the residues of M2 necessary for LC3 interaction. However, some of the data showing the consequence of the cleavage for viral replication could be better clarified.

We thank Reviewer 3 for their kind comments and we propose further experiments to clarify the consequences of cleavage.

Major comments:

- In Fig3A-B, the authors seek to demonstrate that the localization of M2 to the plasma membrane requires LIR motif. However, the representative images for cell infected with the delta 86-97 mutant show relatively few cells are expressing M2 raising questions of the infectivity of this mutant virus or if the overall expression of M2 in this assay is less for the delta 86-97 mutant. The authors should consider first quantifying the ratio of M2 cell surface staining over total M2 staining and second re-evaluate the representative images chosen.

Revision Plan

We will include more examples of permeabilised cells in which comparable numbers of cells are M2 positive between mutants. We will also include high-content microscopy based quantification to support this. To clarify, we confirm that the quantification of M2 intensity in the plasma membrane is carried out relative to the number of M2 positive cells, as the reviewer agrees is the most accurate way. To avoid confusion, we will update figure legends to describe more accurately the quantification process. A comparison between surface M2 and total M2 cannot be done on an individual cell basis, as once cells are permeabilized (to look for internal M2), robust differentiation between surface and internal M2 is difficult. The above clarification and additional data should provide the necessary support for our conclusions.

- In fig3E, it is unclear what is being quantified in the graph as the legend and text lines 222-223 mention that spot intensity was measured but the y axis indicates LC3 relocalization intensity. Given LC3 is punctated particularly in the cytosol, it is unclear which spots of LC3 they are referring to. Based on the images shown, using a graph with LC3 surface staining as performed for M2 would clarify the data. The authors should clarify the reporting of these data in the results section. Additionally, the images of the control non-infected cells should be added to 3C.

We agree with the reviewer on this point. The figure will be updated to describe more accurately what is being quantified. Additionally, images for uninfected cells in 3C will be added.

- The data in Fig4 and FigS3 need to be strengthened to be conclusive. The volcano plot in FigS3A indicates that there is more LC3B and IAV proteins in M2 D85A than M2delta86-97. However in Fig4E, both LC3 I and LC3 II are increased in virions M2 delta 86-97 compared to M2 D85A which is opposite to the authors' conclusions in lines 244-245. In other words, the total amount of lipidated LC3 is higher in virions from IAV M2 without LIR motif than M2 with LIR. LC3II/I ratio in fig4F would suggest in virions containing M2 with LIR motif, LC3B II may be preferentially incorporated compared to virions containing M2 without LIR, which incorporates both LC3B I and LC3B II. Since this is a critical point made by the authors, performing a co-immunoprecipitation of M2 D58A and M2delta86-97 in the particles and then assessing for binding of LC3 I or II would bolster their conclusions.

Figure 4F quantifies the ratio of LC3II to LC3I in infectious particles. **Another two repeats used to quantify this ratio will be shown in addition, with a better representation of increased amounts of lipidated LC3II in M2^{D85A} infectious particles, as well as an increased LC3II/LC3I ration in said particles when compared to M2^{Δ86-97}.** Because of the low yield acquired from the purification of IAV virions, performing an IP would be difficult. Even if this were technically feasible it would not prove that M2 is binding LC3 inside the virion – we do not make this claim in our paper, merely that LC3B can be detected in the purified viral particles. We will clarify this point in the revised manuscript.

- In Fig4J, even if statistically significant, the PFU difference between M2 D85A and M2 delta86-97 is

Revision Plan

minimal, performing growth curve assay would help appreciate this difference over time. In Thp1 cells, as the authors show caspase cleavage of M2 at time point 12h 14h 16hpi etc... (fig1), they should also show PFU data at these same time points for M2 mutant D85A compared to WT and M2 delta 86-97.

Minor comments:

We agree with the reviewer and indeed this was a point we attempted to make in our manuscript: Figure 4J shows a statistically significant difference between the titers. However, in the text we state that, even though statistically significant, the difference is much smaller than in other titer quantifications performed. Given the nature of a plaque assay, differences of less than a log fold cannot be considered as definitively indicating biological significance. We will clarify this in a revised manuscript. **We will also provide the relevant growth kinetics** (as per response to Reviewer 2).

- The title of Fig4 and FigS3 and in text line 226 should be changed as M2 incorporation into virions is not shown and not described in the text. Plus, in figS3B, the authors show that between the M2 mutants, there is no difference in the abundance of M2 and other viral proteins compared to M1.

The title of Figures 4 and S3 will be changed to more accurately reflect all of the points made by the figure.

- In the image shown in Fig4H the number of plaques is higher for M2delta86-97 even though the size is smaller than M2 WT. Could the authors clarify in the text of the results section how they quantify PFU in their plaque assay and if they used a size criterion when quantifying the number of plaques?

The images of plaques are taken at different dilutions, with the M2^{Δ86-97} image belonging to two dilutions lower than the M2^{WT} image. **We will include the calculation used for PFU/mL, which does not take into account plaque size. Furthermore, images of the whole plate, showing plaqued serial dilutions will be shown.**

- In fig3B, the legend indicates 8 hpi but on the graphs it is 9 hpi.

We thank the reviewer for pointing out this mistake. Both should read 8 hpi, this will be corrected in the new manuscript.

Reviewer #3 (Significance (Required)):

The authors demonstrated that IAV M2 binding to LC3 is regulated by caspase cleavage. The authors clearly identify the cleavage site and the caspase involved: caspase 6. The cleaved form of M2 seems relevant to IAV infection as it is accumulating after 12hpi. Using a M2 mutant D85A that

Revision Plan

cannot be cleaved by caspase 6 and truncated M2 mutant delta86-97 mimicking caspase cleaved M2, the authors are able to elegantly address the role of M2 cleavage. However, the importance of M2 caspase cleavage on IAV infection is not demonstrated.

Eventually, addressing the impact of the caspase cleavage of M2 LIR motif on autophagy or CASM would be interesting.

- Advance: conceptual.

- Audience: basic research, specialized in virology, specialized in autophagy.

- Field of expertise: virology, autophagy.

We agree with the reviewer that we have made a conceptual advance in our understanding of the cell biology of influenza A virus infection. We have also determined the structure of the terminal part of the M2 tail in complex with LC3B. The biological importance of the phenotypes we show are most likely in transmission of the virus between hosts, which for IAV would require animal experiments outside the scope of this study. We have demonstrated regulation of the LIR motif by caspase cleavage in a variety of ways, using cell biological and biochemical methods. IAV is a very significant human and animal pathogen, and we believe we have made an important advance in describing a host-pathogen interaction of relevance for viral egress.

Dear Dr. Beale,

Thank you for the transfer of your research manuscript from Review Commons to EMBO reports. I now went through your manuscript, the referee reports from Review Commons (attached again below) and your revision plan. The referees have several comments, concerns, and suggestions to improve the manuscript, indicating that a major revision of the manuscript is necessary to allow publication of the study.

Going through your revision plan, it seems that the referee points will be adequately addressed during revision. I thus invite you to revise your manuscript accordingly with the understanding that all concerns must be addressed in the revised manuscript and/or in a final detailed point-by-point response (as indicated in your revision plan). Acceptance of your manuscript will depend on a positive outcome of another round of review using the same set of referees.

It is EMBO reports policy to allow a single round of major revision only and acceptance of the manuscript will therefore depend on the completeness of your responses included in the next, final version of the manuscript.

- 1) a .docx formatted version of the final manuscript text (including legends for main figures, EV figures and tables), but without the figures included. Figure legends should be compiled at the end of the manuscript text.
- 2) individual production quality figure files as .eps, .tif, .jpg (one file per figure), of main figures (up to 8) and EV figures (up to 5). Please upload these as separate, individual files upon re-submission.

- 3) a final .docx formatted letter INCLUDING the reviewers' reports and your detailed point-by-point responses to their comments. As part of the EMBO Press transparent editorial process, the point-by-point response is part of the Review Process File (RPF), which will be published alongside your paper.

- 4) a complete author checklist, which you can download from our author guidelines (<https://www.embopress.org/page/journal/14693178/authorguide>). Please insert page numbers in the checklist to indicate where the requested information can be found in the manuscript. The completed author checklist will also be part of the RPF.

- 5) that primary datasets produced in this study (e.g. RNA-seq, ChIP-seq, structural and array data) are deposited in an

appropriate public database. If no primary datasets have been deposited, please also state this in a dedicated section (e.g. 'No primary datasets have been generated and deposited'), see below.

The accession numbers and database should be listed in a formal "Data Availability" section (placed after Materials & Methods) that follows the model below. This is now mandatory (like the COI statement). Please note that the Data Availability Section is restricted to new primary data that are part of this study. This section is mandatory. As indicated above, if no primary datasets have been deposited, please state this in this section

Data availability

8) Regarding data quantification and statistics, please make sure that the number "n" for how many independent experiments were performed, their nature (biological versus technical replicates), the bars and error bars (e.g. SEM, SD) and the test used to calculate p-values is indicated in the respective figure legends (also for potential EV figures and all those in the final Appendix). Please also check that all the p-values are explained in the legend, and that these fit to those shown in the figure. Please provide statistical testing where applicable. Please avoid the phrase 'independent experiment', but clearly state if these were biological or technical replicates. Please also indicate (e.g. with n.s.) if testing was performed, but the differences are not significant. In case n=2, please show the data as separate datapoints without error bars and statistics. See also: <http://www.embopress.org/page/journal/14693178/authorguide#statisticalanalysis>

9) Please also note our reference format:

10) We updated our journal's competing interests policy in January 2022 and request authors to consider both actual and perceived competing interests. Please review the policy <https://www.embopress.org/competing-interests> and update your competing interests if necessary. Please name this section 'Disclosure and Competing Interests Statement' and put it after the Acknowledgements section.

11) We now use CRediT to specify the contributions of each author in the journal submission system. CRediT replaces the author contribution section. Please use the free text box to provide more detailed descriptions and do NOT provide an author contributions section in the revised manuscript text file. See also guide to authors:

<https://www.embopress.org/page/journal/14693178/authorguide#authorshippinguidelines>

12) Please add scale bars of similar style and thickness to all the microscopic images, using clearly visible black or white bars (depending on the background). Please place these in the lower right corner of the images themselves. Please do not write on or near the bars in the image but define the size in the respective figure legend.

13) Please make sure that all the funding information is also entered into the online submission system and that it is complete

and similar to the one in the acknowledgement section of the manuscript text file.

14) All Materials and Methods need to be described in the main text using our 'Structured Methods' format, which is required for all research articles. According to this format, the Materials and Methods section should include a Reagents and Tools Table (listing key reagents, experimental models, software and relevant equipment and including their sources and relevant identifiers), uploaded as separate file, followed by a Methods and Protocols section in which we encourage the authors to describe their methods using a step-by-step protocol format with bullet points, to facilitate the adoption of the methodologies across labs. More information on how to adhere to this format as well as downloadable templates (.doc or .xls) for the Reagents and Tools Table can be found in our author guidelines (section 'Structured Methods'):

15) Please add up to five keywords to the manuscript and order the manuscript sections like this, using these names: Title page - Abstract - Keywords - Introduction - Results - Discussion - Methods - Data availability section - Acknowledgements - Disclosure and Competing Interests Statement - References - Figure legends - Expanded View Figure legends

Please note that all corresponding authors are required to supply an ORCID ID for their name upon submission of a revised manuscript. Please find instructions on how to link the ORCID ID to the account in our manuscript tracking system in our Author guidelines: <http://www.embopress.org/page/journal/14693178/authorguide#authorshipguidelines>

I look forward to seeing a revised version of your manuscript when it is ready. Please let me know if you have questions or comments regarding the revision.

Best,

Achim Breiling
Senior editor
EMBO reports

Referee #1:

The Matrix 2 (M2) protein of influenza A virus (IAV) is a single pass transmembrane protein known to act as a tetrameric ion channel that is important for both viral entry and egress. The paper by Figueras-Nova et al. entitled "Caspase cleavage of Influenza A virus M2 disrupts M2-LC3 interaction and regulates virion production" reports on the regulation of IAV virion production through a regulatory interplay between a caspase cleavage site and a LC3 interacting region (LIR) motif in M2. In its C-terminal cytoplasmic tail the IAV M2 protein contains a C-terminal LIR motif interacting with LC3. The authors show that this LIR motif is preceded by a functional caspase cleavage motif cleaved predominantly by caspase-6, with some contribution from caspase-3: The motif 82-SAVD-85 directs cleavage after the aspartate (D) at position 85. The cleavage leads to loss of the remaining C terminal sequence from amino acid 86 to 97. The core LIR motif 91-FVSI-94 LIR motif is then lost from M2 which can no longer bind LC3. As previously described by the same group using point mutations in the LIR motif (Ref 12.), loss of a functional LIR., here by caspase- mediated deletion of the LIR, affects the virion production and inhibits filamentous budding. LC3B lipidation is increased upon treatment with a caspase inhibitor. The authors show for the first time that LC3 is included into IAV virions via binding to M2. Furthermore, they also report a co-crystal structure of the M2 C terminus (aa 70-97), containing the caspase cleavage site and LIR, and LC3B (aa 3-125) adding new insights into this interaction and showing that the caspase cleavage site is in a flexible region N-terminal to the LIR. This work shows how caspase cleavage may modulate LC3B lipidation, trafficking to the plasma membrane, incorporation of LC3B in the virions, filamentous budding and virion production (viral titer).

****Major comments:****

The findings reported here are very well supported by the data shown. This is a very clearly written paper with well described

and nicely visualized results that are accompanied by adequate statistical analyses.

The authors report a new way the LC3B binding to the C-terminal tail of the M2 proteins is regulated and suggest that this is an adaptation the virus has made to adjust virion production to host cell status by hijacking the function of host caspases. They show that the caspase cleavage motif is evolutionary conserved and use that as an argument. Perhaps it could be discussed if it also could be an argument that the host protects itself against a too massive virion production as this could be too detrimental to the host? Would it not also be an evolutionary advantage to the virus in the long run by avoiding killing the host?

A question I may raise which is optional as it may be too much work to address as part of this study is if the reported regulation of LC3B binding has any role in regulating the ion channel function of the M2 tetramer?

****Minor comments:****

Delete "with" in line 145.

Line 217: It should be written more specifically how "cells were surface stained with M2"

In the Introduction a description of what filamentous vs "spherical" budding is, could perhaps be included as I missed that reading through, although it comes in the end of the Discussion.

****Significance:****

This is a very well performed study with a sound experimental strategy and well performed assays with clear results increasing our insight into the interplay between the Influenza A virus and host cells. Although caspase mediated cleavage of the autophagy receptor and signaling scaffold protein p62 (Ref. 25), removing the LIR and LC3-binding, has been reported before I consider this study as novel in reporting this type of regulation of LC3 binding. The cleavage of p62 deletes a large part of the protein while here it is a "clean" deletion of the LIR sequence representing a conceptual advance of regulation of LC3 binding.

The study also reports for the first time on LC3B incorporated into virions.

The effects on trafficking to the plasma membrane and viral budding and virion production are similar to those reported before (Ref. 12) using viruses with point mutations crippling the LIR motif.

This research will be of interested to all studying virus- host interaction and to the autophagy field both as a non autophagic role of LC3B, and as a regulatory mechanism of LIR-LC3B interactions involving the irreversible caspase cleavage-mediated deletion of the LIR motif.

Referee #2:

The influenza A virus (IAV) M2 protein is small transmembrane protein which plays a role in virus entry and egress. In a previous study, Beale et al. (2014) identified an LC3-interacting region (LIR) in the M2 cytoplasmic domain that was found to recruit the LC3B protein to the plasma membrane. Recombinant IAV harboring mutations in the LIR motif showed reduced particle stability and lost filamentous morphology.

In the present study, Figueras-Novoa et al. show that the LIR motif is removed in response to activation of cellular caspases. The authors demonstrate that in IAV-infected THP-1 cells M2 is partially cleaved at the motif (82)SAVD(85) A by caspase 6. Caspase inhibitors abolished cleavage, and a mutant virus harboring the D85A substitution was found to be resistant to caspase action. A crystal structure of purified M2 C- terminus and LC3B revealed that the caspase cleavage site lies in a flexible region that is accessible to caspases.

Mutant virus encoding a truncated M2 protein (M2 86-97) was unable to interact with LC3, in accordance with the absence of the LIR motif. The M2 86-97 mutant showed reduced lipidation of LC3, while enhanced lipidation of LC3 was observed when wild-type virus-infected cells were treated with caspase inhibitors. The authors also observed that cell surface transport of M2 86-97 but not M2-D85A was impaired. However, in purified virus particles a mix of cleaved and uncleaved M2 was detected. The authors also demonstrated that lipidated LC3B was present in purified virions of wild-type virus particles but even more abundant in M2-D85A virions. Finally, M2 86-97 mutants produced significantly less infectious particles compared to wild-type virus while the D85A cleavage mutant replicated to similar titers than wt virus.

Based on these findings the authors concluded that caspases regulate the interaction of M2 protein with LC3 which impacts virion production. Specifically, they propose that caspase-mediated removal of the LIR motif may enable a switch between filamentous and non-filamentous budding in response to depletion of cellular resources. However, the authors were unable to rescue a filamentous IAV with a truncated M2 protein and therefore could not provide direct proof for their guess.

While the data are sound and presented well, they do not support the conclusions of the authors.

1. To the authors opinion, the conserved caspase cleavage site in the M2 protein might provide an evolutionary advantage for

the virus. However, the M2-D85A mutation has no effect on viral replication, so the biological significance of why M2 needs to be cleaved at all is unclear. The conclusion that caspase-induced M2 cleavage is a fine-tuning mechanism of IAV has not been supported by experiments.

2. The finding that the permanently truncated IAV M2 mutant virus was substantially attenuated does not necessarily mean that abrogation of M2-LC3 interaction was responsible for this attenuation. As the M2 protein plays a role in virus budding at the plasma membrane (recruitment of M1 protein, induction of membrane curvature, membrane scission), the impaired transport of the truncated M2 protein might already explain that the virus was attenuated and that incorporation of the protein into the viral envelope was reduced.

3. It is also not clear whether the loss of the C-terminal 11 amino acids may have affected the interaction of the M2 protein with other proteins such as TRAPPC6A-delta (Zhu et al., 2017).

4. The authors did not rule out whether the truncation of the M2 protein by 11 amino acids would have an effect on proton channel activity. Proton channel activity, however, might be important to preserve the metastable conformation of HA in the secretory pathway and might be also important for virus uncoating.

5. M2 86-97 induced less LC3 lipidation than wild-type M2 or the D85A mutant. The remaining lipidation was attributed to the ion channel activity of the M2 protein. Can the authors rule out that the truncation of the M2 protein led to reduced ion channel activity which in turn led to reduced LC3B lipidation?

6. The suggested role of caspase cleavage as a regulatory switch between filamentous and spherical virions (lines 304- 313) is highly speculative as long as the authors do not provide any experimental proof for it. The authors indicated that they were unable to rescue filamentous IAV with M2 86-97. However, would it be possible to use caspase inhibitors to test their hypothesis?

7. The authors used only the PR8 strain for their studies, a highly cell culture-adapted strain with spherical morphology. Are the findings obtained with this strain are also valid for others IAV strains?

8. The authors mainly used the THP-1 cells for their studies, a human macrophage-like cell line. However, human IAV mostly replicate in epithelial cells of the respiratory tract and cause only abortive infections of macrophages. Why did the authors choose this cell line? Can the findings obtained with this cell line be translated to epithelial cells of the airways?

****Minor issues:****

- Fig. 1C: There seem to be quite some differences in the cleavage efficiency of M2 between panels A, B, C, and D? Any explanations?

- Fig. 1: Panel E: The labeling of the first amino acids as aa 76 seems to be wrong!

- Line 147: ...caspase mediated disruption of the M2-LC3 interaction (Fig 2A-B). Should be Fig. 2A-C.

- Growth kinetics of the various mutant viruses are missing?

- Line 195: The authors speculate that aa85 is important for viral fitness: That should be demonstrated!

****Significance:****

Authors concluded that caspases regulate the interaction of M2 protein with LC3 which impacts virion production. Specifically, they propose that caspase-mediated removal of the LIR motif may enable a switch between filamentous and non-filamentous budding in response to depletion of cellular resources. However, the authors were unable to rescue a filamentous IAV with a truncated M2 protein and therefore could not provide direct proof for their guess.

Referee #3:

In this article, the authors identify a caspase cleavage site in the influenza A virus (IAV) Matrix 2 protein (M2) that leads to a truncated form of M2 deleted from its C-term LC3-interacting region (LIR). This cleaved form of M2 is seen and accumulates starting at 12 hours post-infection. IAV expressing M2 delta 86-97 mutant, corresponding to cleaved M2, seems to disrupt LC3B localization to cell plasma membrane upon infection. The authors also show that the IAV M2 delta 86-97 has a reduced viral titer compared to IAV WT. Overall the data are quite exciting where the authors identify the specific caspase responsible for the cleavage and show the residues of M2 necessary for LC3 interaction. However, some of the data showing the consequence of the cleavage for viral replication could be better clarified.

****Major comments:****

- In Fig3A-B, the authors seek to demonstrate that the localization of M2 to the plasma membrane requires LIR motif. However, the representative images for cell infected with the delta 86-97 mutant show relatively few cell are expressing M2 raising questions of the infectivity of this mutant virus or if the overall expression of M2 in this assay is less for the delta 86-97 mutant. The authors should consider first quantifying the ratio of M2 cell surface staining over total M2 staining and second re-evaluate the representative images chosen.

- In fig3E, it is unclear what is being quantified in the graph as the legend and text lines 222-223 mention that spot intensity was measured but the y axis indicates LC3 relocalization intensity. Given LC3 is punctated particularly in the cytosol, It is unclear which spots of LC3 they are referring to. Based on the images shown, using a graph with LC3 surface staining as performed for M2 would clarify the data. The authors should clarify the reporting of these data in the results section. Additionally, the images

of the control non-infected cells should be added to 3C.

- The data in Fig4 and FigS3 need to be strengthened to be conclusive. The volcano plot in FigS3A indicates that there is more LC3B and IAV proteins in M2 D85A than M2delta86-97. However in Fig4E, both LC3 I and LC3 II are increased in virions M2 delta 86-97 compared to M2 D85A which is opposite to the authors' conclusions in lines 244-245. In other words, the total amount of lipidated LC3 is higher in virions from IAV M2 without LIR motif than M2 with LIR. LC3II/I ratio in fig4F would suggest in virions containing M2 with LIR motif, LC3B II may be preferentially incorporated compared to virions containing M2 without LIR, which incorporates both LC3B I and LC3B II. Since this is a critical point made by the authors, performing a co-immunoprecipitation of M2 D58A and M2delta86-97 in the particles and then assessing for binding of LC3 I or II would bolster their conclusions.

- In Fig4J, even if statistically significant, the PFU difference between M2 D85A and M2 delta86-97 is minimal, performing growth curve assay would help appreciate this difference over time. In Thp1 cells, as the authors show caspase cleavage of M2 at time point 12h 14h 16hpi etc... (fig1), they should also show PFU data at these same time points for M2 mutant D85A compared to WT and M2 delta 86-97.

****Minor comments:****

- The title of Fig4 and FigS3 and in text line 226 should be changed as M2 incorporation into virions is not shown and not described in the text. Plus, in figS3B, the authors show that between the M2 mutants, there is no difference in the abundance of M2 and other viral proteins compared to M1.

- In the image shown in Fig4H the number of plaques is higher for M2delta86-97 even though the size is smaller than M2 WT. Could the authors clarify in the text of the results section how they quantify PFU in their plaque assay and if they used a size criterion when quantifying the number of plaques?

- In fig3B, the legend indicates 8 hpi but on the graphs it is 9 hpi.

****Significance:****

The authors demonstrated that IAV M2 binding to LC3 is regulated by caspase cleavage. The authors clearly identify the cleavage site and the caspase involved: caspase 6. The cleaved form of M2 seems relevant to IAV infection as it is accumulating after 12hpi. Using a M2 mutant D85A that cannot be cleaved by caspase 6 and truncated M2 mutant delta86-97 mimicking caspase cleaved M2, the authors are able to elegantly address the role of M2 cleavage. However, the importance of M2 caspase cleavage on IAV infection is not demonstrated.

Eventually, addressing the impact of the caspase cleavage of M2 LIR motif on autophagy or CASM would be interesting.

- ***Advance:*** conceptual.

- ***Audience:*** basic research, specialized in virology, specialized in autophagy.

- ***Field of expertise:*** virology, autophagy.

We thank the reviewers for their positive assessment of our manuscript. We have performed additional experiments that have been suggested by the reviewers and have enhanced our study. We have added to this document the additional data included in the manuscript, and where changes have been made in the text, we have highlighted the lines changed.

Reviewer #1 (Evidence, reproducibility and clarity (Required)):

1. EVIDENCE, REPRODUCIBILITY AND CLARITY

Summary:

The Matrix 2 (M2) protein of influenza A virus (IAV) is a single pass transmembrane protein known to act as a tetrameric ion channel that is important for both viral entry and egress. The paper by Figueras-Nova et al. entitled "Caspase cleavage of Influenza A virus M2 disrupts M2-LC3 interaction and regulates virion production" reports on the regulation of IAV virion production through a regulatory interplay between a caspase cleavage site and a LC3 interacting region (LIR) motif in M2. In its C-terminal cytoplasmic tail the IAV M2 protein contains a C-terminal LIR motif interacting with LC3. The authors show that this LIR motif is preceded by a functional caspase cleavage motif cleaved predominantly by caspase-6, with some contribution from caspase-3: The motif 82-SAVD-85 directs cleavage after the aspartate (D) at position 85. The cleavage leads to loss of the remaining C terminal sequence from amino acid 86 to 97. The core LIR motif 91-FVSI-94 LIR motif is then lost from M2 which can no longer bind LC3. As previously described by the same group using point mutations in the LIR motif (Ref 12.), loss of a functional LIR., here by caspase-mediated deletion of the LIR, affects the virion production and inhibits filamentous budding. LC3B lipidation is increased upon treatment with a caspase inhibitor. The authors show for the first time that LC3 is included into IAV virions via binding to M2. Furthermore, they also report a co-crystal structure of the M2 C terminus (aa 70-97), containing the caspase cleavage site and LIR, and LC3B (aa 3-125) adding new insights into this interaction and showing that the caspase cleavage site is in a flexible region N-terminal to the LIR. This work shows how caspase cleavage may modulate LC3B lipidation, trafficking to the plasma membrane, incorporation of LC3B in the virions, filamentous budding and virion production (viral titer).

Major comments:

The findings reported here are very well supported by the data shown. This is a very clearly written paper with well described and nicely visualized results that are accompanied by adequate statistical analyses.

We thank the reviewer for their assessment of our manuscript.

The authors report a new way the LC3B binding to the C-terminal tail of the M2 proteins is regulated and suggest that this is an adaptation the virus has made to adjust virion production to host cell status by hijacking the function of host caspases. They show that the caspase cleavage motif is evolutionary conserved and use that as an argument. Perhaps it could be discussed if it also could be an argument that the host protects itself against a too massive virion production as this could be too detrimental to the host? Would it not also be an evolutionary advantage to the virus in the long run by avoiding killing the host?

This is an interesting point. We agree there could be advantage for the virus not to overproduce virions under certain circumstances. Consistent with this, caspase-6 deficient mice had increased mortality in response to IAV PR8 infection, and exhibited higher viral titres in their lungs (Zheng, 2021; doi: [10.1016/j.cell.2020.03.040](https://doi.org/10.1016/j.cell.2020.03.040)). This is also relevant for the comments made by Reviewer 2. The manuscript has been updated to include a discussion of this point (lines 399-402).

A question I may raise which is optional as it may be too much work to address as part of this study is if the reported regulation of LC3B binding has any role in regulating the ion channel function of the M2 tetramer?

It is well established that there is no impact of distal C-terminal truncations on M2 ion channel activity (Cady et al., 2009, doi: [10.1021/bi9008837](https://doi.org/10.1021/bi9008837) Schnell and Chou, doi: [10.1038/nature06531](https://doi.org/10.1038/nature06531); Nguyen et al., 2008, doi: [10.1021/bi801315m](https://doi.org/10.1021/bi801315m); Tobler et al., 1999, doi: [10.1128/jvi.73.12.9695-9701.1999](https://doi.org/10.1128/jvi.73.12.9695-9701.1999)). This is also consistent with data from our lab (Ulferts et al., 2021, doi: [10.1016/j.celrep.2021.109899](https://doi.org/10.1016/j.celrep.2021.109899), Beale et al., 2014, doi: [10.1016/j.chom.2014.01.006](https://doi.org/10.1016/j.chom.2014.01.006)) as well as others (Ren et al., 2015, doi: [10.1128/JVI.00576-15](https://doi.org/10.1128/JVI.00576-15)) showing the effects of the LIR motif and the proton channel are distinct. We appreciate the reviewer suggesting further work here as optional, but there is already compelling evidence to show there is no substantial effect of the LIR motif on ion channel activity. (See also Reviewer 2 points 4 and 5).

Minor comments:

Delete "with" in line 145.

This has been changed in the updated manuscript (line 180).

Line 217: It should be written more specifically how "cells were surface stained with M2"

Surface staining of M2 is carried out in unpermeabilised cells. The sentence has been updated to clarify this (lines 259-260).

In the Introduction a description of what filamentous vs "spherical" budding is, could perhaps be included as I missed that reading through, although it comes in the end of the Discussion.

A description of the pleomorphic nature of IAV has been added to the Introduction (lines 47-54).

Reviewer #1 (Significance (Required)):

2. SIGNIFICANCE

This is a very well performed study with a sound experimental strategy and well performed assays with clear results increasing our insight into the interplay between the Influenza A virus and host cells. Although caspase mediated cleavage of the autophagy receptor and signaling scaffold protein p62 (Ref. 25), removing the LIR and LC3-binding, has been reported before I consider this study as novel in reporting

this type of regulation of LC3 binding. The cleavage of p62 deletes a large part of the protein while here it is a "clean" deletion of the LIR sequence representing a conceptual advance of regulation of LC3 binding.

The study also reports for the first time on LC3B incorporated into virions.

The effects on trafficking to the plasma membrane and viral budding and virion production are similar to those reported before (Ref. 12) using viruses with point mutations crippling the LIR motif.

This research will be of interested to all studying virus- host interaction and to the autophagy field both as a non autophagic role of LC3B, and as a regulatory mechanism of LIR-LC3B interactions involving the irreversible caspase cleavage-mediated deletion of the LIR motif.

We thank the reviewer for this assessment of our manuscript.

Reviewer #2 (Evidence, reproducibility and clarity (Required)):

The influenza A virus (IAV) M2 protein is small transmembrane protein which plays a role in virus entry and egress. In a previous study, Beale et al. (2014) identified an LC3-interacting region (LIR) in the M2 cytoplasmic domain that was found to recruit the LC3B protein to the plasma membrane. Recombinant IAV harboring mutations in the LIR motif showed reduced particle stability and lost filamentous morphology.

In the present study, Figueras-Novoa et al. show that the LIR motif is removed in response to activation of cellular caspases. The authors demonstrate that in IAV-infected THP-1 cells M2 is partially cleaved at the motif (82)SAVD(85)↓A by caspase 6. Caspase inhibitors abolished cleavage, and a mutant virus harboring the D85A substitution was found to be resistant to caspase action. A crystal structure of purified M2 C- terminus and LC3B revealed that the caspase cleavage site lies in a flexible region that is accessible to caspases.

Mutant virus encoding a truncated M2 protein (M2 Δ 86-97) was unable to interact with LC3, in accordance with the absence of the LIR motif. The M2 Δ 86-97 mutant showed reduced lipidation of LC3, while enhanced lipidation of LC3 was observed when wild-type virus-infected cells were treated with caspase inhibitors. The authors also observed that cell surface transport of M2 Δ 86-97 but not M2-D85A was impaired. However, in purified virus particles a mix of cleaved and uncleaved M2 was detected. The authors also demonstrated that lipidated LC3B was present in purified virions of wild-type virus particles but even more abundant in M2-D85A virions. Finally, M2 Δ 86-97 mutants produced significantly less infectious particles compared to wild-type virus while the D85A cleavage mutant replicated to similar titers than wt virus.

Based on these findings the authors concluded that caspases regulate the interaction of M2 protein with LC3 which impacts virion production. Specifically, they propose that caspase-mediated removal of the LIR motif may enable a switch between filamentous and non-filamentous budding in response to depletion of cellular resources. However, the authors were unable to rescue a filamentous IAV with a truncated M2 protein and therefore could not provide direct proof for their guess.

While the data are sound and presented well, they do not support the conclusions of the authors.

1. To the authors opinion, the conserved caspase cleavage site in the M2 protein might provide an evolutionary advantage for the virus. However, the M2-D85A mutation has no effect on viral replication, so the biological significance of why M2 needs to be cleaved at all is unclear. The conclusion that caspase-induced M2 cleavage is a fine-tuning mechanism of IAV has not been supported by experiments.

We thank the reviewer for the assessment of our data. We think the reviewer is specifically objecting to the phrase “We conclude that this highly conserved interaction and cleavage act as a regulatory mechanism exploited by IAV to fine-tune virion production in different cellular contexts.” This is a reasonable inference from our results, especially now since we have been able to rescue an M2 truncation mutant on a filamentous background, but we accept that it is not proven. We have changed the wording to make this clear (line 371).

2. The finding that the permanently truncated IAV M2 mutant virus was substantially attenuated does not necessarily mean that abrogation of M2-LC3 interaction was responsible for this attenuation. As the M2 protein plays a role in virus budding at the plasma membrane (recruitment of M1 protein, induction of membrane curvature, membrane scission), the impaired transport of the truncated M2 protein might already explain that the virus was attenuated and that incorporation of the protein into the viral envelope was reduced.

We have confirmed this further with additional experiments using LIR mutants. We have recapitulated the plasma membrane transport defect of truncated M2 with LIR mutants including the newly characterised M2^{D87A} and M2^{D88A} mutants (Figure S3F, 266-273). Furthermore, infection with a complete LIR motif mutant (FVSI_AAAA substitution) recapitulated the M2^{Δ86-97} mutant phenotype in M2 plasma membrane staining (Figure S3B, lines 235-243). This strongly implies that the truncation mutant phenotype is due to the lack of LIR motif.

3. It is also not clear whether the loss of the C-terminal 11 amino acids may have affected the interaction of the M2 protein with other proteins such as TRAPPC6A-delta (Zhu et al., 2017).

This is a reasonable point, however Zhu et al., 2017 (<https://doi.org/10.1128/jvi.01757-16>) reported that the interaction with TRAPPC6A retains M2 intracellularly. If the phenotype observed with our truncation was due to the loss of interaction with TRAPPC6A, the opposite phenotype would be expected (more M2 in the plasma membrane with the truncated M2^{Δ86-97} mutant). To address this directly we generated another M2 mutant virus that has disrupted the reported TRAPPC6A binding site (M2^{EL-AA}) and assessed M2 plasma membrane localization (Figure S3C, lines 243-248). This indicates that in our system, and at the specific time point used, the interaction between TRAPPC6A and M2 does not impact M2 plasma membrane localization.

4. The authors did not rule out whether the truncation of the M2 protein by 11 amino acids would have an effect on proton channel activity. Proton channel activity, however, might be important to preserve the metastable conformation of HA in the secretory pathway and might be also important for virus uncoating.

5. M2^{Δ86-97} induced less LC3 lipidation than wild-type M2 or the D85A mutant. The remaining lipidation was attributed to the ion channel activity of the M2 protein. Can the authors rule out that the truncation of the M2 protein led to reduced ion channel activity which in turn led to reduced LC3B lipidation?

We have addressed points 4 and 5 in response to Reviewer 1.

6. The suggested role of caspase cleavage as a regulatory switch between filamentous and spherical virions (lines 304- 313) is highly speculative as long as the authors do not provide any experimental proof for it. The authors indicated that they were unable to rescue filamentous IAV with M2^{Δ86-97}. However, would it be possible to use caspase inhibitors to test their hypothesis?

The use of caspase inhibitors would only increase the amount of full length M2 present, and does not provide an alternative strategy for increasing the proportion of truncated M2. To address this point we designed an alternative strategy to rescue

the M2^{Δ86-97} mutant in a filamentous background. HEK293T and MDCK cells were generated to express Udorn M2 upon doxycycline treatment. This complex trans-complementation approach enabled us to generate a M2^{Δ86-97} mutant in the MUd background. As expected, the M2^{Δ86-97} mutant is defective in filamentous budding (lines 324-359). In Figure 6 we show that infection with the M2^{Δ86-97} mutant leads to a substantial reduction in the generation of filament bundles, both through immunofluorescence and scanning electron microscopy. Figure S5 provides information on the generation of the M2^{Δ86-97} mutant and verification by sequencing.

A**C****B**

agtgctgtggatgctgacgacagtcattttgtcagcatagagctGGAGTAA
 82 83 84 85 86 87 88 89 90 91 92 93 94 95 96 97
 S A V D A D D S H F V S I E L E *
 812-1078

template sequence pUual UOorn segment 7

7. The authors used only the PR8 strain for their studies, a highly cell culture-adapted strain with spherical morphology. Are the findings obtained with this strain are also valid for others IAV strains?

As we highlight in Figure 2I, both the caspase cleavage motif and LIR motif are highly conserved in human IAV strains. PR8 was used as it is the reverse genetic system in use and approved for use in the lab. We have addressed this by obtaining cell lysates infected with clinical isolates of seasonal IAV strains in circulation prior to extensive passage in cell culture. These were available to us as a byproduct of the clinically directed activities of the Worldwide Influenza Centre (WIC). They replicate to varying extents in cell culture and are variably cytopathic, so we cannot provide the same level of standardisation as with a laboratory adapted strain. Nonetheless we could demonstrate that all of the strains tested exhibit M2 cleavage to varying extents (Figure 1G-H, lines 144-158). This indicates that M2 cleavage occurs in

seasonal IAV strains and is neither a peculiarity of the PR8 strain nor an adaptation to cell culture.

8. The authors mainly used the THP-1 cells for their studies, a human macrophage-like cell line. However, human IAV mostly replicate in epithelial cells of the respiratory tract and cause only abortive infections of macrophages. Why did the authors choose this cell line? Can the findings obtained with this cell line be translated to epithelial cells of the airways?

THP-1 cells are widely used for the study of caspase activity. However, we also show M2 cleavage in MDCK cells and HAP1 cells. PR8 infection of A549 cells does not induce significant amounts of cell death in the infection time points used and, as caspase activation is linked to cell death, we did not observe substantial M2 cleavage in this cell type. Infection with seasonal H3N2 viruses above was carried out in MDCK cells, which indicates cleavage also occurs in epithelial cells.

9. Minor issues:

- Fig. 1C: There seem to be quite some differences in the cleavage efficiency of M2 between panels A, B, C, and D? Any explanations?

Different cell types (THP-1 cells and HAP1 cells) are used for the experiments mentioned above, which largely accounts for the different amount of M2 cleavage.

- Fig. 1: Panel E: The labeling of the first amino acids as aa 76 seems to be wrong!

We thank the reviewer for pointing this out, this has been corrected in the updated manuscript.

- Line 147: ...caspase mediated disruption of the M2-LC3 interaction (Fig 2A-B). Should be Fig. 2A-C.

This sentence was referring to Figure 2A-B, as it refers to LC3B lipidation and not the colP. The reference to the figures has been moved to reflect the intended meaning (line 171).

- Growth kinetics of the various mutant viruses are missing?

We have provided growth kinetics for the relevant mutants ($M2^{D85A}$ and $M2^{\Delta 86-97}$) (Figure 5D, lines 311-316).

- Line 195: The authors speculate that aa85 is important for viral fitness: That should be demonstrated!

This speculation is based on the very strong conservation of D85 in human IAV strains. The importance of D85 in viral fitness (permitting cleavage of M2) is only likely to be directly demonstrable in transmission models (for example ferrets) which is not feasible or justifiable within the scope of this project. As pointed out above, caspase-6 deficient mice exhibited increased mortality in response to IAV PR8 infection, and presented an increase in viral spread in the lungs (Zheng, 2021; doi: 10.1016/j.cell.2020.03.040), which is consistent with inhibition of cleavage leading to a more virulent infection *in vivo*.

Reviewer #2 (Significance (Required)):

Authors concluded that caspases regulate the interaction of M2 protein with LC3 which impacts virion production. Specifically, they propose that caspase-mediated removal of the LIR motif may enable a switch between filamentous and non-filamentous budding in response to depletion of cellular resources. However, the authors were unable to rescue a filamentous IAV with a truncated M2 protein and therefore could not provide direct proof for their guess.

As stated in the response to the comments above, we have rescued the permanently

truncated M2^{Δ86-97} mutant in a MUD background, which has provided further support for our hypothesis (Figure 6).

Reviewer #3 (Evidence, reproducibility and clarity (Required)):

Summary :

In this article, the authors identify a caspase cleavage site in the influenza A virus (IAV) Matrix 2 protein (M2) that leads to a truncated form of M2 deleted from its C-term LC3-interacting region (LIR). This cleaved form of M2 is seen and accumulates starting at 12 hours post-infection. IAV expressing M2 delta 86-97 mutant, corresponding to cleaved M2, seems to disrupt LC3B localization to cell plasma membrane upon infection. The authors also show that the IAV M2 delta 86-97 has a reduced viral titer compared to IAV WT. Overall the data are quite exciting where the authors identify the specific caspase responsible for the cleavage and show the residues of M2 necessary for LC3 interaction. However, some of the data showing the consequence of the cleavage for viral replication could be better clarified.

We thank Reviewer 3 for their kind comments and we have carried out further experiments to clarify the consequences of cleavage.

Major comments:

- In Fig3A-B, the authors seek to demonstrate that the localization of M2 to the plasma membrane requires LIR motif. However, the representative images for cell infected with the delta 86-97 mutant show relatively few cells expressing M2 raising questions of the infectivity of this mutant virus or if the overall expression of M2 in this assay is less for the delta 86-97 mutant. The authors should consider first quantifying the ratio of M2 cell surface staining over total M2 staining and second re-evaluate the representative images chosen.

We have included high-content microscopy based quantification of number of M2 positive cells (Figure S3A, lines 233-235). To clarify, we confirm that the quantification of M2 intensity in the plasma membrane is carried out relative to the number of M2 positive cells, as the reviewer agrees is the most accurate way. To avoid confusion, we have updated figure legends to describe the quantification process more precisely. A comparison between surface M2 and total M2 cannot be done on an individual cell basis, as once cells are permeabilized (to look for internal M2), robust differentiation between surface and internal M2 is not feasible. The above clarification and additional data provides further support for our conclusions.

- In fig3E, it is unclear what is being quantified in the graph as the legend and text lines 222-223 mention that spot intensity was measured but the y axis indicates LC3 relocation intensity. Given LC3 is punctated particularly in the cytosol, It is unclear which spots of LC3 they are referring to. Based on the images shown, using a graph with LC3 surface staining as performed for M2 would clarify the data. The authors should clarify the reporting of these data in the results section. Additionally, the images of the control non-infected cells should be added to 3C.

We agree with the reviewer on this point. The figure has been updated to describe more accurately what is being quantified (Figure 3E, lines 258-266), and an additional quantification of this measurement has been added (Figure S3E, lines 261-266). Additionally, images for uninfected cells in 3C have been added (Figure S3D, lines 255-257).

- The data in Fig4 and FigS3 need to be strengthened to be conclusive. The volcano plot in FigS3A indicates that there is more LC3B and IAV proteins in M2 D85A than M2 Δ 86-97. However in Fig4E, both LC3 I and LC3 II are increased in virions M2 delta 86-97 compared to M2 D85A which is opposite to the authors' conclusions in lines 244-245. In other words, the total amount of lipidated LC3 is higher in virions from IAV M2 without LIR motif than M2 with LIR. LC3II/I ratio in fig4F would suggest in virions containing M2 with LIR motif, LC3B II may be preferentially incorporated compared to virions containing M2 without LIR, which incorporates both LC3B I and LC3B II. Since this is a critical point made by the authors, performing a co-immunoprecipitation of M2 D58A and M2 Δ 86-97 in the particles and then assessing for binding of LC3 I or II would bolster their conclusions.

Figure 4F quantifies the ratio of LC3II to LC3I in infectious particles. Another two repeats used to quantify this ratio have been shown in addition, with a better representation of increased amounts of lipidated LC3-II in M2^{D85A} infectious particles, as well as an increased LC3-II/LC3-I ratio in these particles when compared to M2 ^{Δ 86-97} (Figure S4A, line 294). Because of the low yield acquired from the purification of IAV virions, performing an IP would be difficult. Even if this were technically feasible it would not prove that M2 is binding LC3 inside the virion – we do not make this claim in our paper, merely that LC3B can be detected in the purified viral particle preparations (which has never previously been demonstrated).

- In Fig4J, even if statistically significant, the PFU difference between M2 D85A and M2 delta86-97 is minimal, performing growth curve assay would help appreciate this difference over time. In Thp1 cells, as the authors show caspase cleavage of M2 at time point 12h 14h 16hpi etc... (fig1), they should also show PFU data at these same time points for M2 mutant D85A compared to WT and M2 delta 86-97.

Minor comments:

We agree with the reviewer and indeed this was a point we attempted to make in our manuscript: Figure 5E shows a statistically significant difference between the titers.

However, in the text we state that, even though statistically significant, the difference is much smaller than in other titer quantifications performed. Given the nature of a plaque assay, differences of less than a log fold cannot be considered as definitively indicating biological significance. We have attempted to clarify this in the revised manuscript (lines 316-322). We have also provided the relevant growth kinetics (as per response to Reviewer 2).

- The title of Fig4 and FigS3 and in text line 226 should be changed as M2 incorporation into virions is not shown and not described in the text. Plus, in figS3B, the authors show that between the M2 mutants, there is no difference in the abundance of M2 and other viral proteins compared to M1.

Figure 4 has been divided into Figure 4 (M2 cleavage reduces M2 incorporation into virions) and 5 (IAV M2^{Δ86-97} virus is attenuated) to ease understanding of the data. The reference to M2 incorporation into virions is to Figure 4A-B where incorporation of cleaved and full-length M2 into virions is measured for purified M2^{WT} infectious particles.

- In the image shown in Fig4H the number of plaques is higher for M2delta86-97 even though the size is smaller than M2 WT. Could the authors clarify in the text of the results section how they quantify PFU in their plaque assay and if they used a size criterion when quantifying the number of plaques?

The images of plaques (now Figure 5B) are taken at different dilutions, with the M2^{Δ86-97} image belonging to two dilutions lower than the M2^{WT} image. We have included the calculation used for PFU/mL in methods (lines 595-597), which does not take into account plaque size. Furthermore, images of the whole plate, showing plaqued serial dilutions will be submitted as part of the source data submission.

- In fig3B, the legend indicates 8 hpi but on the graphs it is 9 hpi.

We thank the reviewer for pointing out this mistake. Both should read 8 hpi, this has been corrected in the new manuscript.

Reviewer #3 (Significance (Required)):

The authors demonstrated that IAV M2 binding to LC3 is regulated by caspase cleavage. The authors clearly identify the cleavage site and the caspase involved: caspase 6. The cleaved form of M2 seems relevant to IAV infection as it is accumulating after 12hpi. Using a M2 mutant D85A that cannot be cleaved by caspase 6 and truncated M2 mutant delta86-97 mimicking caspase cleaved M2, the authors are able to elegantly address the role of M2 cleavage. However, the importance of M2 caspase cleavage on IAV infection is not demonstrated. Eventually, addressing the impact of the caspase cleavage of M2 LIR motif on autophagy or CASM would be interesting.

- Advance: conceptual.

- Audience: basic research, specialized in virology, specialized in autophagy.

- Field of expertise: virology, autophagy.

We agree with the reviewer that we have made a conceptual advance in our understanding of the cell biology of influenza infection. We have also determined the structure of the terminal part of the M2 tail in complex with LC3B. The biological importance of the phenotypes we show are most likely in transmission of the virus between hosts, which for IAV would require animal experiments outside the scope of this study. We have demonstrated regulation of the LIR motif by caspase cleavage in a variety of ways, using cell biological and biochemical methods. Additional experiments performed as a part of the review process have shown that cleavage of M2 leads to reduced formation of filament bundles, indicating caspase cleavage may indeed regulate virion formation as hypothesized in our initial submission. IAV is a very significant human and animal pathogen, and we believe we have made an important advance in describing a host-pathogen interaction of relevance for viral egress.

Dear Dr. Beale,

Thank you for the submission of your revised manuscript to our editorial offices. I have now received the reports from the referees that I asked to re-evaluate the study, you will find below. As you will see, all three referees now support publication of the study in EMBO reports. However, referee #1 has some remaining concerns and suggestions to improve the manuscript I ask you to address in a final revised manuscript/

Moreover, I have these editorial requests

- Please provide individual production quality figure files as .eps, .tif, .jpg (one file per figure), also of the Supplementary figures. The Expanded View format, which will be displayed in the main HTML of the paper in a collapsible format, has replaced the Supplementary information. Thus, please name these figures Figure EV1, Figure EV2 etc. and upload these as separate, individual files upon re-submission. Moreover, Please update their callouts and their naming in their legends.

- Please add up to 5 keywords to the manuscript and put these below the abstract.

- Please order the sections like this using these names:

Title page - Abstract - Keywords - Introduction - Results - Discussion - Methods - Data availability section (DAS) - Acknowledgements (including funding information) - Disclosure and Competing Interests Statement - References - Figure legends - Expanded View Figure legends

- We now use CRediT to specify the contributions of each author in the journal submission system. CRediT replaces the author contribution section. Please use the free text box to provide more detailed descriptions and do NOT provide your final manuscript text file with an author contributions section. See also our guide to authors: <https://www.embopress.org/page/journal/14693178/authorguide#authorshipguidelines>

- We updated our journal's competing interests policy in January 2022 and request authors to consider both actual and perceived competing interests. Please review the policy <https://www.embopress.org/competing-interests> and add a statement declaring your competing interests. Please name that section 'Disclosure and Competing Interests Statement' and add it after the author contributions section.

- Please upload a complete author checklist with your revised manuscript, which you can download from our author guidelines (<https://www.embopress.org/page/journal/14693178/authorguide>). Please insert page numbers in the checklist to indicate where the requested information can be found in the manuscript. The completed author checklist will also be part of the RPF.

- Please add scale bars of similar style and thickness to all microscopic images, using clearly visible black or white bars (depending on the background). Please place these in the lower right corner of the images themselves. Please do not write on or near the bars in the image but define the size in the respective figure legend.

- Please check again that the number "n" for how many independent experiments were performed, their nature (biological versus technical replicates), the bars and error bars (e.g. SEM, SD) and the test used to calculate p-values is indicated in the respective figure legends (main, EV and Appendix figures). Please also check that all the p-values are explained in the legend, and that these fit to those shown in the figure. Please provide statistical testing where applicable. Please avoid the phrase 'independent experiment', but clearly state if these were biological or technical replicates. Please also indicate (e.g. with n.s.) if testing was performed, but the differences are not significant. In case n=2, please show the data as separate datapoints without error bars and statistics (see Fig S1H/EV1H). See also: <http://www.embopress.org/page/journal/14693178/authorguide#statisticalanalysis>

If n<5, please show single datapoints for diagrams. Moreover:

- Please note that information related to n is missing in the legends of supplementary figures 4b-c.

- Although 'n' is provided, please describe the nature of entity for 'n' in the legends of figures 2b, e; 4f; 5a, c-e, supplementary figures 1a, c-d, f.

- Please note that the error bars are not defined in the legend of supplementary figure 4c.

- Please note that the exact p values are not provided in the legends of figures 1c-d; 2b, e; 3b, d-e; 4f; 5a, c-e, supplementary figures 1a, c-d, f; 3b-c, e; 4d.

- Please indicate the statistical test used for data analysis in the legend of supplementary figure 4b.

- All materials and methods used need to be described in the main text using our 'Structured Methods' format, which is required for all research articles. According to this format, the Methods section should include a Reagents and Tools Table (listing key

reagents, experimental models, software, and relevant equipment and including their sources and relevant identifiers), uploaded as separate file, followed by a Methods section in which we encourage the authors to describe their methods using a step-by-step protocol format with bullet points, to facilitate the adoption of the methodologies across labs. More information on how to adhere to this format as well as downloadable templates (.doc or .xls) for the Reagents and Tools Table can be found in our author guidelines (section 'Structured Methods'):

- Please make sure that all the funding information is also entered into the online submission system and is complete and similar to the one in the manuscript text file (in the Acknowledgements). Presently, the grants from Cancer Research UK and the Medical Research Council, the Wellcome Trust and MRC (MR/T00004X/1) are missing from the submission system. Please check.

- Please use our reference format:

- There is a Supplementary Table S1 and a Supplementary table 1 called out in the text, but there is no such file uploaded. Please provide this, either as EV Table (Table EV1), as dataset (Dataset EV1) or as part of the reagents & tools table. Finally, please adjust the respective callouts.

- Thank you for providing the requested source data. Please upload this as one folder per figure (with all files for one figure in one folder and ZIPped together).

In addition, I would need from you uploaded separately:

- a short, two-sentence summary of the manuscript (not more than 35 words).

- two to four short (!) bullet points highlighting the key findings of your study (two lines each).

- a schematic summary figure as separate file that provides a sketch of the major findings (not a data image) in jpeg or tiff format (with the exact width of 550 pixels and a height of not more than 400 pixels) that can be used as a visual synopsis on our website.

Best,

Referee #1:

The manuscript «Caspase cleavage of influenza A virus M2 disrupts M2-LC3 interaction and regulates virion production" by Figueras-Novoa has been revised and additional experimental data have been added. The data are sound and support the notion that caspase 6 cleaves the M2 cytoplasmic domain at the sequence motif 82SAVD85, which results in impaired M2 cell surface transport and virus production. Still, the conclusions drawn by the authors are not always supported by the data and should be formulated more cautiously. In addition, the discussion should address some more of these aspects.

1. Lines 253-254: "Therefore, we conclude the M2 LIR motif regulates M2 plasma membrane transport". This conclusion is based on the transport defects observed with the M2 86-97 and M2(FVSI_AAAA) mutants. However, the truncation of the M2 cytosolic domain by 11 amino acids might have had a general impact on M2 conformation and transport which was independent of the LIR motif. Likewise, the replacement of the "FVSI" motif by four alanine residues does not prove that the LIR motif "FXXI" regulates M2 plasma membrane transport. Substitution of the aromatic phenylalanine residue by a single alanine residue might have been more appropriate to test the importance of this motif. The conclusion might be modified in the way that the last 11 amino acids are important for M2 cell surface transport!

2. Lines 278-279: "The results suggest that the ability of M2 to interact with LC3B affects M2 transport to the plasma membrane". This conclusion could be further supported by data that M2(D87A) and M2(D88A) mutants influence binding to LC3B as described in Figure 2C for the M2-86-97 mutant.

3. Lines 365-367: "These data confirmed the role of the M2 LIR motif in filament formation, and support a role of caspase cleavage in the regulation of IAV virion morphology". This conclusion is not supported by the data. The authors show that

filamentous virus production is impaired in recombinant virus encoding the Udorn M segment and a truncated M2. However, release of spherical PR8 virions encoding truncated M2 protein is also impaired (see for example Fig 5A). However, the budding defect of this virus might not be easily visualized by scanning electron microscopy. The conclusion should be modified in the way that the C-terminal 11 amino acids of the M2 protein are important for release of both spherical and filamentous virions.

4. The authors did not investigate the impact of the truncation of the M2 cytoplasmic tail by 11 amino acids for M2 ion channel activity but rather referred to previous reports (see response to reviewer 1). To my opinion, the authors should include their arguments and the literature into the Discussion!

5. Lines 380-382: The authors hypothesize that "this highly conserved interaction and cleavage may act as a regulatory mechanism exploited by IAV to fine-tune virion production in different cellular contexts". To my opinion, the importance of the caspase cleavage site is overestimated since the M2(D85A) mutant (which is not cleaved by caspase 6) does not show a phenotype such as increased virus production.

Referee #2:

This is a very nice study which also is strengthened by the new data added in the revision process. In my view the study is clearly acceptable for publication.

Referee #3:

Influenza A virus (IAV) M2 protein has previously been shown to interact with LC3 through its LIR domain. In this paper the authors identify an M2 cleavage site targeted by caspase 6 (with some contribution from caspase 3) leading to the removal of the LIR domain. To bolster these data, the authors provide a crystal structure of C-term M2 including the LIR domain and LC3 demonstrating that the caspase cleavage site is outside the LIR and in a flexible region that could feasibly interact with the caspase. The authors then compare in different assays virus expressing either wild type M2, a non-cleavable M2 D85A, and a truncation mutant M2delta86-97 lacking the LIR domain. Through these assays, they demonstrate that LC3 lipidation and plasma membrane targeting of LC3 and M2 depend on the LIR domain. Given that IAV is an enveloped virus that steals the host plasma membrane during budding, the authors show that lipidated LC3 is preferentially included in mature virions from M2-D85A compared to the delta86-97 mutant. The identification of the cleavage site, caspase responsible for cleavage and LC3 incorporation into the virions are notable advancements to understanding host-IAV interaction and an unconventional function of an autophagy-associated protein.

On the original manuscript the authors presented some data to show that the loss of the LIR impacted viral replication, but the data needed additional growth curves to fully support this conclusion. The authors had also suggested that the cleavage was important for the morphological transition of the virions between filamentous and non-filamentous budding but failed rescue experiments impeded this important conclusion.

In the revised manuscript, there is now adequate data to support these two important claims that highlights the significance of LC3 cleavage in the context of IAV replication. The authors have included a new growth curve that shows a ~2 log decrease in titer of the M2delta86-97 virus compared to WT or D85A. They successfully performed complementation experiments in a filamentous A/Udorn/72 background to test if the presence of the LIR domain impacted filament formation. The Mud_delat86-89 failed to form filaments suggesting that the LIR domain played a role in the filamentation of IAV.

Given these significant improvements made on revision, the revised manuscript is appropriate for publication without any additional revision.

1. Lines 253-254: "Therefore, we conclude the M2 LIR motif regulates M2 plasma membrane transport". This conclusion is based on the transport defects observed with the M2 Δ 86-97 and M2(FVSI_AAAA) mutants. However, the truncation of the M2 cytosolic domain by 11 amino acids might have had a general impact on M2 conformation and transport which was independent of the LIR motif. Likewise, the replacement of the "FVSI" motif by four alanine residues does not prove that the LIR motif "FXXI" regulates M2 plasma membrane transport. Substitution of the aromatic phenylalanine residue by a single alanine residue might have been more appropriate to test the importance of this motif. The conclusion might be modified in the way that the last 11 amino acids are important for M2 cell surface transport!

We have updated the sentence indicated by the reviewer in lines 263-264, which now reads:

Therefore, we conclude the last 11 amino acids of the M2 cytoplasmic tail regulate M2 plasma membrane transport.

2. Lines 278-279: "The results suggest that the ability of M2 to interact with LC3B affects M2 transport to the plasma membrane". This conclusion could be further supported by data that M2(D87A) and M2(D88A) mutants influence binding to LC3B as described in Figure 2C for the M2-86-97 mutant.

We previously measured the monomeric wild-type LIR motif K_d for LC3B at approximately 10 μ M (Beale et al., 2014). The truncated mutant lacks the entire LIR motif and so the failure to interact with LC3 is evident on a pull-down experiment. More subtle effects are not straightforward to demonstrate with these semi-quantitative techniques. Whilst it would in principle be possible to measure reduced affinities of D87 and D88 mutants by performing careful biophysical measurements we do not think this would add much to our understanding of the M2/LC3 interaction. It is established that acidic residues in similar positions contribute to affinity in many LC3-LIR interactions. We think it is reasonable to infer from the crystal structure of M2 and LC3B in complex and decreased GFP-LC3B puncta in D87A and D88A infection that D87A and D88A mutations affect M2-LC3B binding. To provide further clarification on this in the text, lines 286-291 have been modified as follows:

However, infection with M2^{D87A} and M2^{D88A} resulted in decreased GFP-LC3B relocalisation, consistent with the importance of these residues for M2-LC3B binding (Figure EV3F). In addition, reduced M2 surface staining was observed with the M2^{D87A} and M2^{D88A} relative to M2^{WT} (Figure EV3F), suggesting that the ability of M2 to interact with LC3B affects M2 transport to the plasma membrane.

3. Lines 365-367: "These data confirmed the role of the M2 LIR motif in filament formation, and support a role of caspase cleavage in the regulation of IAV virion morphology". This conclusion is not supported by the data. The authors show that filamentous virus production is impaired in recombinant virus encoding the Udorn M segment and a truncated M2. However, release of spherical PR8 virions encoding truncated M2 protein is also impaired (see for example Fig 5A). However, the budding defect of this virus might not be easily visualized by scanning electron microscopy. The conclusion should be modified in the way that the C-terminal 11

amino acids of the M2 protein are important for release of both spherical and filamentous virions.

We have edited the sentence indicated by the reviewer in lines 376-379 to be:

These data confirmed the role of the last 11 amino acids of the M2 C-terminal tail for both spherical virion production and filament formation, which suggests a role for caspase cleavage in the regulation of IAV morphology.

4. The authors did not investigate the impact of the truncation of the M2 cytoplasmic tail by 11 amino acids for M2 ion channel activity but rather referred to previous reports (see response to reviewer 1). To my opinion, the authors should include their arguments and the literature into the Discussion!

We agree with the reviewer that we should integrate the relevant literature in the Discussion. We have added this point in lines 437-442, and rearranged the last paragraph of the Discussion section to ease understanding. The passage reads:

While many of the roles carried out by M2 are associated with its ion channel activity, this is not required for filament formation (Rossman et al, 2010). This is consistent with previous reports indicating distal C-terminal tail truncations have no significant impact on M2 ion channel activity (Cady et al, 2009, Schnell & Chou, 2008, Nguyen et al, 2008, Tobler et al, 1999).

5. Lines 380-382: The authors hypothesize that "this highly conserved interaction and cleavage may act as a regulatory mechanism exploited by IAV to fine-tune virion production in different cellular contexts". To my opinion, the importance of the caspase cleavage site is overestimated since the M2(D85A) mutant (which is not cleaved by caspase 6) does not show a phenotype such as increased virus production.

We agree that there are some unanswered questions about the lack of increased viral fitness of D85A in cell culture models. We suggest that this likely reflects the fact that IAV M2^{WT} infection already produces an excess of full length M2 than required for maximal virion production (i.e. full length M2 is not a limiting factor), and therefore any additional full length M2 present in IAV M2^{D85A} infection has no impact on titer.

Consistent with this, caspase-6 deficient mice had increased mortality in response to IAV PR8 infection, and exhibited higher viral titres in their lungs (Zheng, 2021; doi: [10.1016/j.cell.2020.03.040](https://doi.org/10.1016/j.cell.2020.03.040)). We refer to this paper in lines 418-421.

We have added this to the discussion, in lines 423-425, as follows:

While we did not observe an increase in titer with the M2^{D85A} mutant relative to M2^{WT} infection, this may reflect a non rate-limiting abundance of full length M2 in the cell culture models of infection used in this study.

Dr. Rupert Beale
Francis Crick Institute
United Kingdom

Dear Dr. Beale,

Thank you for the submission of your final revised manuscript. Going through your p-b-p-response, I consider the remaining points of referee #1 as adequately addressed.

I am thus very pleased to accept your manuscript for publication in the next available issue of EMBO reports. Thank you for your contribution to our journal.

Yours sincerely,

Rev_Com_number: RC-2024-02467

New_manu_number: EMBOR-2024-59903V3

Corr_author: Beale

Title: Caspase cleavage of Influenza A virus M2 disrupts M2-LC3 interaction and regulates virion production